# BINDING MODE MATTERS: RESIDUE-GUIDED DRUG DISCOVERY VIA EXPLORATIVE PREFERENCES

## ABSTRACT

The discovery of novel hit or lead molecules requires navigating a vast chemical space to identify compounds with optimal binding modes, which are typically unknown beforehand. Despite various generative approaches, they have predominantly relied on optimizing a monolithic scalar docking score to guide generation, masking the distinct contributions of key binding determinants. In this work, we introduce a paradigm shift by formulating target-based drug design as a multi-objective exploration task, where each objective explicitly corresponds to enhancing interactions with a specific key residue. To this end, we introduce **BindMol**, a novel generative framework that integrates a fragment-based generator with a customized multi-objective reinforcement learning algorithm. By incorporating explorative preferences during training, our approach efficiently uncovers molecules with distinct and desirable binding profiles. Empirical evaluations demonstrate that **BindMol** facilitates the discovery of structurally novel, high-affinity compounds across five protein targets and establishes new state-of-the-art records on the multi-property optimization tasks in GuacaMol benchmarks, thereby providing a versatile paradigm for goal-directed drug discovery.

## 1 INTRODUCTION

The advent of deep generative modeling is reshaping the paradigm of drug design, offering a powerful toolkit that spans *de novo discovery* and *goal-oriented generation* (Du et al., 2024). Rather than learning from a generalized distribution, goal-oriented generation imposes explicit objectives to restrict the chemical space for sampling. To this end, reinforcement learning (RL) stands out as a natural solution to chemical space navigation, demonstrating great potential in conditional generation (Olivecrona et al., 2017; Lee et al., 2024) and optimization (Zhou et al., 2019; Yang et al., 2021).

The reward function, a critical component in RL, provides the training signal that guides policy model training and ultimately governs task performance. Despite its importance, most RL-based approaches still rely on simplified and coarse-grained reward schemes, such as docking scores for binding affinity optimization. However, such scalar measures often overlook the complexity of interactions: *compounds targeting the same protein can adopt different binding modes*. For instance, two well-known GLP-1 agonists exhibit similar affinities but engage distinct sub-regions, resulting in either broad-spectrum or biased signaling (Zhang et al., 2020), as illustrated in Figure 1.

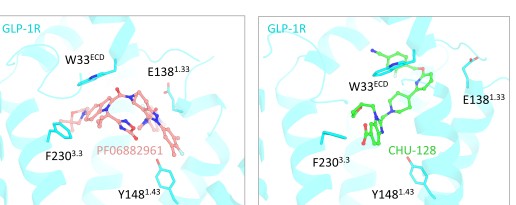

Figure 1: Illustration of two well-known agonists of GLP-1R, highlighting their distinct binding modes within the same protein pocket.

To address this limitation, we formulate target-based drug discovery as a multi-objective optimization problem, with each objective corresponding to a key residue within the binding pocket. This granular reward structure not only promotes exploration of diverse binding modes but also effectively circumvents the issues of reward hacking (Amodei et al., 2016) and local optima inherent to monolithic scoring functions (Fromer & Coley, 2023). As a result, it is expected to provide a richer pool of compound candidates, especially in binding modes, significantly enhancing the prospect of identifying inhibitors with high selectivity and druggability.

Specifically, we propose a novel reinforcement learning framework for drug discovery, termed **BindMol**, which departs from data-driven distribution learning and instead enables explicitly explorative molecule generation within a carefully structured action space guided by task-specific rewards. Our framework is built upon three tightly coupled components: a chemically grounded action space, a multi-objective RL algorithm, and a biologically informed reward design.

To ensure the chemical validity and synthesizability of generated molecules, we ground our exploration in a fragment-based action space (Yang et al., 2021). By assembling molecules in a *Lego-like* fashion, this design inherently strictly enforces pharmacochemical validity (Kong et al., 2022; Zhu et al., 2023) and avoids the generation of unrealistic polycyclic systems often observed in atom-by-atom approaches (Guan et al., 2024). To enable precise control over multi-objective trade-offs, we establish a principled actor-critic framework, *Envelope SAC*, that achieves consistent joint optimization of preference-aware critics and policies. This formulation advances the theoretical scope of the envelope operator (Yang et al., 2019) beyond its value-based origins by rigorously aligning vectorized Bellman updates with stochastic policy gradients. Thereby, it enables efficient approximation of the Pareto frontier across diverse preferences. For the reward formulation, we move beyond simple scalar scores and construct a vectorized reward landscape. Utilizing interaction profiling tools (Salentin et al., 2015), we compute residue-level interactions to guide the agent toward specific binding modes. This is complemented by a global docking score to mitigate reward sparsity, ensuring the agent captures both specific interaction patterns and overall binding affinity.

We extensively validate the effectiveness of our approach across multiple tasks designed to reflect realistic drug discovery scenarios. On binding affinity optimization for five protein targets, our method identified hit candidates with greater novelty, higher binding affinity, and diverse interaction modes compared to existing approaches. Furthermore, on seven multi-property optimization tasks from the GuacaMol benchmark (Brown et al., 2019), **BindMol** achieves state-of-the-art performance compared to 12 generative baselines, underscoring its efficacy in multi-objective drug rediscovery. The main contributions of this work are three-fold:

❶ We propose the first molecular generation framework that treats binding-mode diversity as a *first-class* modeling objective, explicitly accounting for biologically grounded structural variations. To support this, we pioneer the integration of vectorized multi-objective preference signals directly within the RL framework, introducing a new paradigm for controllable generation that avoids the limitations of scalarization heuristics.

❷ We propose *Envelope SAC*, a unified multi-objective RL algorithm that extends the envelope Q-learning operator to the actor-critic setting. This formulation addresses the non-trivial challenge of jointly optimizing multi-preference-aware critics and actors, establishing a robust mechanism for exploring the binding modes in high-dimensional chemical spaces.

❸ We demonstrate the superiority of **BindMol** across diverse benchmarks. It outperforms existing methods in generating high-affinity ligands with novel binding modes across five protein targets and achieves state-of-the-art results on the comprehensive GuacaMol multi-objective benchmark.

## 2  RELATED WORK

**Multi-objective reinforcement learning**  Existing work in multi-objective reinforcement learning (MORL) can be broadly categorized into tabular and deep approaches, each comprising both single-policy and multi-policy methods (Felten et al., 2023). Tabular single-policy methods typically optimize a scalarized objective (Van Moffaert et al., 2013), whereas tabular multi-policy algorithms learn collections of policies under varying preference weights or directly maintain sets of non-dominated value estimates—such as in Pareto Q-learning (Van Moffaert & Nowé, 2014)—to approximate the Pareto front. In the deep setting, single-policy methods like EUPG (Roijers et al., 2018) focus on optimizing user-specified utilities, commonly under the Expected Scalarized Return formulation. Deep multi-policy methods constitute the largest group, spanning weight-conditioned architectures (Yang et al., 2019; Lu et al., 2023), evolutionary population-based techniques (Xu et al., 2020), return-conditioned models (Reymond et al., 2022), and frameworks leveraging Generalized Policy Improvement (Alegre et al., 2023). Despite their differences, these methods share the goal of efficiently capturing a diverse set of trade-offs and primarily vary in how they scalarize objectives, incorporate preference information, and approximate the Pareto-optimal policy set.

**RL-driven molecular generation** Deep reinforcement learning has been widely applied to molecular design with diverse strategies. Sequence-based policy tuning methods, such as REINVENT (Olivecrona et al., 2017) and its variants using curriculum learning (Guo et al., 2022) or augmented memory (Guo & Schwaller, 2024), adapt SMILES generators toward desired chemical properties. Graph-based approaches (Zhou et al., 2019) directly optimize molecules through chemistry-aware edit actions without pretraining. On the other hand, fragment-based methods assemble molecules from fragments while addressing exploration and chemical validity (Yang et al., 2021; Lee et al., 2024). More recently, structure-aware techniques such as AliDiff (Gu et al., 2024) and DecompDPO (Cheng et al., 2024) align generative models with protein-binding preferences, improving target-specific design. Collectively, these works highlight the versatility of RL in drug discovery, though challenges remain in balancing reward hacking and exploration efficiency.

**Fragment-based molecular generation** Recent advances in deep generative modeling have enabled fragment-based drug discovery through various strategies. Fragment-level generation methods, including MARS (Xie et al., 2021), FREED (Yang et al., 2021), and MoLeR (Maziarz et al., 2021), combine optimization techniques with fragment assembly to explore chemical space efficiently. Other approaches leverage informative substructures, such as principal subgraph mining (Kong et al., 2022) or molecular rationales (Jin et al., 2020b), to guide multi-objective molecule generation. Incorporating protein structural context has also proven effective, with models learning transferable subpocket prototypes (Zhang & Liu, 2023) or applying pocket-aware fragment assembly via diffusion (Ghorbani et al., 2023). Beyond generation, optimization frameworks such as HN-GFN (Zhu et al., 2023) enable sample-efficient multi-objective search. Recent advances in antibiotic design (Krishnan et al., 2025) further highlight the practical impact of fragment-based generative strategies in identifying novel compounds with *in vivo* efficacy against drug-resistant pathogens.

# 3 Preliminaries

**Markov decision processes.** We formalize reinforcement learning problems as a Markov Decision Process (MDP), defined by the tuple

$$\mathcal{M} = \langle \mathcal{S}, \mathcal{A}, \mathcal{P}, r \rangle,$$

where $\mathcal{S}$ denotes the state space, $\mathcal{A}$ denotes the action space, $\mathcal{P}(s' \mid s, a)$ denotes the transition probability distribution, and $r(s, a)$ stands for the reward function. For a policy $\pi$, the action-value function $Q^\pi(s, a)$ satisfies the Bellman expectation equation, and the corresponding Bellman operator is given by

$$(TQ)(s, a) := r(s, a) + \gamma \mathbb{E}_{s' \sim \mathcal{P}(\cdot|s,a), \, a' \sim \pi(\cdot|s')} \big[ Q(s', a') \big], \tag{1}$$

where $\gamma \in [0, 1)$ is the discount factor. Building on this principle, the SAC algorithm (Haarnoja et al., 2018) augments the Bellman operator with entropy regularization, so that the agent is encouraged to maximize both return and policy entropy. Formally, the soft Bellman operator is defined as

$$(\tilde{T}Q)(s, a) := r(s, a) + \gamma \mathbb{E}_{s' \sim \mathcal{P}(\cdot|s,a), \, a' \sim \pi(\cdot|s')} \big[ Q(s', a') - \alpha \log \pi(a' \mid s') \big]. \tag{2}$$

**Multi-objective optimization.** We extend this formulation to reinforcement learning with multiple objectives, which can be modeled as a *multi-objective Markov decision process* (MOMDP). Formally, it can be represented by the tuple

$$\mathcal{M} = \langle \mathcal{S}, \mathcal{A}, \mathcal{P}, \mathbf{r}, \Omega \rangle,$$

where the state space $\mathcal{S}$, action space $\mathcal{A}$, and transition kernel $\mathcal{P}$ are defined as in the standard MDP, but the reward function is extended to be vector-valued, $\mathbf{r}(s, a) \in \mathbb{R}^m$. In addition, $\Omega$ denotes the preference space. When $\omega$ is fixed, the MOMDP reduces to a standard single-objective MDP.

# 4 Method

As with generic reinforcement learning pipelines, the `BindMol` framework is divided into three key components: action space, reward function, and training algorithm. In the following subsections, we first introduce the fragment-based generator in Section 4.1, which defines the action space. Then, we describe the proposed multi-objective RL algorithm with explorative preferences in Section 4.2. Finally, we describe the reward function design that perceives the ligand-protein interactions, enabling residue-level controllability in Section 4.3. The framework of `BindMol` is illustrated in Figure 2.

Figure 2: Overview of the **BindMol** consisting of three components: fragment-based generator, Envelope SAC, and residue-level reward design. **(Top left)** Fragment-based generation sequentially attaches fragments from a fragment vocabulary to a partial graph until termination. **(Bottom left)** Overview of the Envelope SAC workflow (double $Q_\theta$ models omitted for clarity). **(Top right)** Residue-level rewards are computed from the sum of interaction counts and docking scores, followed by normalization. **(Bottom right)** Illustration of envelope updates: unlike scalarized RL without maximization over preference, our method identifies the optimal utility for each preference $\omega$.

## 4.1 FRAGMENT-BASED GENERATOR

Following Yang et al. (2021), we frame the molecular generation as a sequential decision process. At time step $t$, the state is the partially built graph $\mathcal{G}_t$. Each transition is achieved by a triplet action

$$\mathbf{a}_t = (u_t, f_t, v_t),$$

where $u_t \in \mathcal{U}(\mathcal{G}_t)$ is an attachment site in the current graph, $f_t \in \mathcal{S}$ is a fragment drawn from the vocabulary $\mathcal{S}$, and $v_t \in \mathcal{V}(f_t)$ is an attachment site on the chosen fragment. Attaching $f_t$ to $\mathcal{G}_t$ at $(u_t, v_t)$ yields $\mathcal{G}_{t+1}$.

Formally, we define three action components governed by sub-policies:

$$\mathbb{P}_1(u_t \mid \mathcal{G}_t) = \Pi_1\big(\mathcal{H}(\mathcal{G}_t), \mathcal{H}_{\text{site}}(\mathcal{G}_t)\big), \tag{3}$$

$$\mathbb{P}_2(f_t \mid u_t, \mathcal{G}_t) = \Pi_2\big(\phi(u_t), \Phi(\mathcal{S})\big), \tag{4}$$

$$\mathbb{P}_3(v_t \mid f_t, u_t, \mathcal{G}_t) = \Pi_3\big(\psi(f_t), \psi_{\text{site}}(f_t)\big), \tag{5}$$

where $\mathcal{H}(\mathcal{G}_t) \in \mathbb{R}^d$ is a global embedding of the partial graph, $\mathcal{H}_{\text{site}}(\mathcal{G}_t) = \{h_u : u \in \mathcal{U}(\mathcal{G}_t)\}$ gives embeddings of candidate sites, and $\phi(u_t) := h_{u_t}$ denotes the embedding of the chosen site. The fragment vocabulary is represented by $\Phi(\mathcal{S}) = \{\phi_b : b \in \mathcal{S}\}$, a collection of precomputed fragment descriptors. Each fragment $f$ is further encoded as a global embedding $\psi(f) \in \mathbb{R}^d$ and its attachment sites as $\psi_{\text{site}}(f) = \{g_v^{(f)} : v \in \mathcal{V}(f)\}$.

Here $\Pi_1, \Pi_2, \Pi_3$ are neural scoring functions that map their inputs to logits; applying a softmax yields categorical distributions $\mathbb{P}_1, \mathbb{P}_2, \mathbb{P}_3$ over the respective action spaces. The overall policy can be factorized as

$$\pi(\mathbf{a}_t \mid \mathcal{G}_t) = \mathbb{P}_1(u_t \mid \mathcal{G}_t)\, \mathbb{P}_2(f_t \mid u_t, \mathcal{G}_t)\, \mathbb{P}_3(v_t \mid f_t, u_t, \mathcal{G}_t). \tag{6}$$

Inspired by Lee et al. (2024), we introduce new fragments by applying graph genetic operations (mutation and crossover) on generated molecules (Jensen, 2019) and incorporating novel substructures back into $\mathcal{S}$. When the vocabulary grows too large, only the top-ranked fragments are retained. This dynamic update ensures continuous exploration beyond the static fragment set. Please refer to Section A for a detailed description of the update mechanism of the dynamic vocabulary.

## 4.2 MULTI-OBJECTIVE RL WITH ENVELOPE UPDATES

Since the optimal preference for residue-based multi-objective optimization is unknown a *priori*, it is necessary to design multi-objective reinforcement learning algorithms with stronger preference exploration in order to discover novel compounds and binding patterns. In this section, we introduce a novel multi-objective SAC algorithm termed *Envelope SAC*. The central idea is to integrate the

envelope-based update mechanism (Yang et al., 2019) into actor–critic learning, thereby enabling stable optimization of vector-valued returns under diverse preferences. This integration ensures that both critics and actors consistently account for the envelope structure, facilitating efficient exploration of trade-offs across objectives without the limitations of fixed linear scalarization.

**Envelope Q-functions.** For simplicity, we first present the general definition of the Q-function here and omit the subscripts of different instantiations used in SAC. The detailed critic parameterization is defined in the subsequent sections. We consider vectorized action-value functions of the form

$$\boldsymbol{Q} : \mathcal{S} \times \mathcal{A} \times \Omega \to \mathbb{R}^m, \quad (s, a, \boldsymbol{\omega}) \mapsto \boldsymbol{Q}(s, a, \boldsymbol{\omega}).$$

For a fixed preference $\boldsymbol{\omega}$, the scalarized utility is defined as

$$Q_{\boldsymbol{\omega}}(s, a) := \boldsymbol{\omega}^\top \boldsymbol{Q}(s, a, \boldsymbol{\omega}). \tag{7}$$

To capture the envelope effect across preferences—meaning to identify the *optimal* possible outcome for a given query preference $\boldsymbol{\omega}$ by considering all available preference views—we first define the *maximizing preference*:

$$\boldsymbol{\omega}^\star(s, a, \boldsymbol{\omega}) \in \arg \max_{\boldsymbol{\omega}' \in \Omega} \boldsymbol{\omega}^\top \boldsymbol{Q}(s, a, \boldsymbol{\omega}'). \tag{8}$$

This $\boldsymbol{\omega}^\star$ selects the preference $\boldsymbol{\omega}'$ that yields the maximal scalarized utility for the query preference $\boldsymbol{\omega}$ at state-action $(s, a)$. In practice, the weight vector $\boldsymbol{\omega}'$ is sampled by first drawing i.i.d. standard normal vectors from $\mathcal{N}(\boldsymbol{0}, \mathbf{I})$, applying an element-wise absolute value to ensure non-negativity, and then performing $\ell_1$ normalization. This ensures that the resulting preferences are uniformly distributed on the probability simplex. With this in place, we now introduce the *soft envelope operator* $\mathcal{H}_\alpha$:

$$(\mathcal{H}_\alpha \boldsymbol{Q})(s, \boldsymbol{\omega}) := \mathbb{E}_{a \sim \pi(\cdot | s, \boldsymbol{\omega})} \Big[ \boldsymbol{Q}(s, a, \boldsymbol{\omega}^\star(s, a, \boldsymbol{\omega})) - \alpha \log \pi(a \mid s, \boldsymbol{\omega}) \Big]. \tag{9}$$

This operator generalizes the envelope filter from multi-objective Q-learning to the entropy-regularized setting. It averages over actions drawn from the current policy, but crucially, it always aligns with the most favorable envelope preference $\boldsymbol{\omega}^\star$ for each observed Q-value, and incorporates the entropy bonus uniformly across all objectives.

**Critic Update.** The critic, parameterized by $\theta$, approximates the vector-valued Q-function $\boldsymbol{Q}_\theta(s, a, \boldsymbol{\omega})$. Its training objective is to minimize the temporal difference error, extending the principles of SAC to our multi-objective envelope setting. Given a transition $(s, a, \boldsymbol{\mathcal{R}}, s')$ with vector reward $\boldsymbol{\mathcal{R}} \in \mathbb{R}^m$, a target network $\boldsymbol{Q}_{\bar{\theta}}$, and a sampled preference $\boldsymbol{\omega}$, the envelope target is defined as:

$$\boldsymbol{y} = \boldsymbol{\mathcal{R}} + \gamma (\mathcal{H}_\alpha \boldsymbol{Q}_{\bar{\theta}})(s', \boldsymbol{\omega}). \tag{10}$$

The critic is trained by minimizing the squared deviation between current prediction and target vector:

$$L_Q(\theta) = \mathbb{E}_{(s,a,\boldsymbol{r},s') \sim \mathcal{D}, \, \boldsymbol{\omega} \sim \mathcal{D}_{\boldsymbol{\omega}}} \Big[ \tfrac{1}{2} \big\| \boldsymbol{Q}_\theta(s, a, \boldsymbol{\omega}) - \boldsymbol{y} \big\|_2^2 \Big]. \tag{11}$$

In contrast to standard SAC, both rewards and targets remain vector-valued. The $\mathcal{H}_\alpha$ operator ensures that each update exploits the most advantageous $(\boldsymbol{\omega}, a)$ pairing observed on the envelope, thereby aligning critic training with the convex hull of attainable utilities.

**Preference-Conditioned Policy Update.** The actor is represented by a stochastic, preference-conditioned policy $\pi_\phi(a \mid s, \boldsymbol{\omega})$. During training, the actor's objective is to maximize the entropy-regularized scalarized utility estimated by the critic. Specifically, the actor loss is formulated to encourage high entropy while maximizing the scalarized envelope Q-value:

$$L_\pi(\phi) = \mathbb{E}_{s \sim \mathcal{D}, \, a \sim \pi_\phi(\cdot | s, \boldsymbol{\omega}), \, \boldsymbol{\omega} \sim \mathcal{D}_{\boldsymbol{\omega}}} \Big[ \alpha \log \pi_\phi(a \mid s, \boldsymbol{\omega}) - \boldsymbol{\omega}^\top \boldsymbol{Q}_\theta(s, a, \boldsymbol{\omega}^\star(s, a, \boldsymbol{\omega})) \Big]. \tag{12}$$

This formulation preserves the entropy-regularized maximum-entropy principle of SAC, while replacing the standard scalar Q-value with the envelope-based utility. This guides the policy towards actions that are optimal considering the trade-offs represented by the envelope.

**Remarks.** Taken together, Equations (10) to (12) establish the learning dynamics of Envelope SAC. The critic approximates vector-valued Q-functions against targets constructed from the soft envelope operator, while the actor maximizes the corresponding scalarized envelope utility under entropy regularization. In implementation, the maximization over $\boldsymbol{\omega}'$ and expectations over actions are approximated by minibatches of sampled preferences and actions. We provide the pseudo-code of Envelope SAC in Algorithm 1.

---

**Algorithm 1:** Envelope Soft Actor-Critic

---

**Input:** Preference sampling distribution $\mathcal{D}_{\boldsymbol{\omega}}$, batch size $B$, number of preferences $N_{\omega}$

Initialize policy $\pi_{\phi}$, critics $\boldsymbol{Q}_{\theta_1,\theta_2}$, target networks $\boldsymbol{Q}_{\bar{\theta}_{1,2}}$, and replay buffer $\mathcal{D}$;

**repeat**

    Interact using $\pi_{\phi}$ with sampled preference $\boldsymbol{\omega}_{\text{env}} \sim \mathcal{D}_{\boldsymbol{\omega}}$, and store transitions in $\mathcal{D}$;

    **for** *each gradient step* **do**

        Sample minibatch $B = \{(s, a, \boldsymbol{r}, s', d)\}$ from $\mathcal{D}$ and preferences $\boldsymbol{\omega}$ from $\mathcal{D}_{\boldsymbol{\omega}}$;

        `// Critic Update`

        Sample next actions $a' \sim \pi_{\phi}(\cdot|s', \boldsymbol{\omega})$;

        Find maximizing preference for the next state:

        $\boldsymbol{\omega}^{\star} \leftarrow \arg\max_{\boldsymbol{\omega} \in W} \left[ \boldsymbol{\omega}^{\top} \left( \min_{i=1,2} \boldsymbol{Q}_{\bar{\theta}_i}(s', a', \boldsymbol{\omega}) \right) \right]$;

        Compute the soft envelope target vector $\boldsymbol{y}$:

        $\boldsymbol{y} \leftarrow \boldsymbol{\mathcal{R}} + \gamma(1 - d) \left( \min_{i=1,2} \boldsymbol{Q}_{\bar{\theta}_i}(s', a', \boldsymbol{\omega}^{\star}) - \alpha \log \pi_{\phi}(a'|s', \boldsymbol{\omega}) \right)$;

        Update critics $\theta_{1,2}$ by minimizing the loss $\mathcal{L}_Q = \frac{1}{2} \sum_{i=1,2} \mathbb{E}_B \left[ \|\boldsymbol{Q}_{\theta_i}(s, a, \boldsymbol{\omega}) - \boldsymbol{y}\|_2^2 \right]$;

        `// Policy Update`

        Find maximizing preference for the current state:

        $\boldsymbol{\omega}^{\star} \leftarrow \arg\max_{\boldsymbol{\omega} \in W} \left[ \boldsymbol{\omega}^{\top} \left( \min_{i=1,2} \boldsymbol{Q}_{\theta_i}(s, a, \boldsymbol{\omega}) \right) \right]$;

        Update policy $\phi$ by minimizing the loss:

        $\mathcal{L}_{\pi} \leftarrow \mathbb{E}_{B, a \sim \pi_{\phi}} \left[ \alpha \log \pi_{\phi}(a|s, \boldsymbol{\omega}) - \boldsymbol{\omega}^{\top} \left( \min_{i=1,2} \boldsymbol{Q}_{\theta_i}(s, a, \boldsymbol{\omega}^{\star}) \right) \right]$;

        `// Target Network Update`

        Update target networks: $\bar{\theta}_i \leftarrow \rho \bar{\theta}_i + (1 - \rho)\theta_i$ for $i = 1, 2$;

**until** *Convergence*;

---

### 4.3 INTERACTION-AWARE REWARD FUNCTION

To encourage the exploration of diverse binding modes, we decompose the conventional binding affinity optimization into residue-level and global docking rewards. Instead of treating all residues in the pocket as potential targets, we first identify a subset of key residues that are most relevant for ligand binding. Specifically, from multiple ligand–protein complexes $\mathcal{C} = \{c_1, \ldots, c_m\}$ obtained from the Protein Data Bank (PDB), we collect the key residue set $R_c$ for each complex as

$$\mathcal{S}_c = \{ s \mid \exists a \in \text{ligand}(c), \ \text{dist}(s, a) \leq 4\text{Å} \},$$

The final residue set is defined either as the union or the intersection over all candidate sets

$$\mathcal{S} = \bigcup / \bigcap_{c \in \mathcal{C}} \mathcal{S}_c.$$

Given a generated ligand, we evaluate its interactions with residues $i \in \mathcal{S}$ using PLIP (Salentin et al., 2015), considering seven common interaction types: *hydrogen bonds, hydrophobic contacts, π–stacking, water bridges, salt bridges, metal complexes, and halogen bonds*. Let $n_t(x, i)$ denote the number of interactions between residue $i$ and ligand $x$ on a specific interaction type $t \in \mathcal{T}$. The raw reward is then normalized to mitigate instability:

$$\hat{n}_t(x, i) = \min\{n_t(x, i), 10\}, \quad \mathcal{R}_t(x, i) = \frac{\hat{n}_t(x, i)}{10} \in [0, 1],$$

where the clipping at 10 prevents overly large values. To address reward sparsity in early training, we combine residue-level rewards with a global docking score. Let $\mathcal{R}_{\text{dock}}(x)$ denote the normalized docking score of ligand $x$. The final reward between ligand $x$ and residue $i$, is defined as

$$\mathcal{R}(x, i) = \frac{1}{|\mathcal{R}_t(x, i)| + 1} \left( \sum_{t \in \mathcal{T}} \mathcal{R}_t(x, i) + \mathcal{R}_{\text{dock}}(x) \right).$$

This formulation provides a fine-grained residue-level feedback while incorporating a global docking reward to stabilize learning when residue-level interactions are initially sparse.

Table 1: **Novel hit ratio (%) results.** The results are the means and the standard deviations of 3 runs. The best results are highlighted in bold.

| Method | Target protein | | | | |
|---|---|---|---|---|---|
| | parp1 | fa7 | 5ht1b | braf | jak2 |
| REINVENT (Olivecrona et al., 2017) | 0.480 (± 0.344) | 0.213 (± 0.081) | 2.453 (± 0.561) | 0.127 (± 0.088) | 0.613 (± 0.167) |
| Graph GA (Jensen, 2019) | 4.811 (± 1.661) | 0.422 (± 0.193) | 7.011 (± 2.732) | 3.767 (± 1.498) | 5.311 (± 1.667) |
| MORLD (Jeon & Kim, 2020) | 0.047 (± 0.050) | 0.007 (± 0.013) | 0.880 (± 0.735) | 0.047 (± 0.040) | 0.227 (± 0.118) |
| HierVAE (Jin et al., 2020a) | 0.553 (± 0.214) | 0.007 (± 0.013) | 0.507 (± 0.278) | 0.207 (± 0.220) | 0.227 (± 0.127) |
| RationaleRL (Jin et al., 2020b) | 4.267 (± 0.450) | 0.900 (± 0.098) | 2.967 (± 0.307) | 0.000 (± 0.000) | 2.967 (± 0.196) |
| FREED (Yang et al., 2021) | 4.627 (± 0.727) | 1.332 (± 0.113) | 16.767 (± 0.897) | 2.940 (± 0.359) | 5.800 (± 0.295) |
| PS-VAE (Kong et al., 2022) | 1.644 (± 0.389) | 0.478 (± 0.140) | 12.622 (± 1.437) | 0.367 (± 0.047) | 4.178 (± 0.933) |
| MOOD (Lee et al., 2023) | 7.017 (± 0.428) | 0.733 (± 0.141) | 18.673 (± 0.423) | 5.240 (± 0.285) | 9.200 (± 0.524) |
| GDSS (Jo et al., 2022) | 1.933 (± 0.208) | 0.368 (± 0.103) | 4.667 (± 0.306) | 0.167 (± 0.134) | 1.167 (± 0.281) |
| RetMol (Wang et al., 2023) | 0.011 (± 0.016) | 0.000 (± 0.000) | 0.033 (± 0.027) | 0.000 (± 0.000) | 0.011 (± 0.016) |
| GEAM (Lee et al., 2024) | 40.635 (± 0.763) | 17.903 (± 1.220) | 36.667 (± 1.929) | 27.887 (± 1.584) | 42.025 (± 1.187) |
| **BindMol** (ours) | **42.538** (± 0.825) | **24.471** (± 0.753) | **41.639** (± 0.142) | **30.867** (± 1.518) | **44.238** (± 0.967) |

Table 2: **Novel top 5% docking score (kcal/mol) results.** The results are the means and the standard deviations of 3 runs. The best results are highlighted in bold.

| Method | Target protein | | | | |
|---|---|---|---|---|---|
| | parp1 | fa7 | 5ht1b | braf | jak2 |
| REINVENT (Olivecrona et al., 2017) | -8.702 (± 0.523) | -7.205 (± 0.264) | -8.770 (± 0.316) | -8.392 (± 0.400) | -8.165 (± 0.277) |
| Graph GA (Jensen, 2019) | -10.949 (± 0.532) | -7.365 (± 0.326) | -10.422 (± 0.670) | -10.789 (± 0.341) | -10.167 (± 0.576) |
| MORLD (Jeon & Kim, 2020) | -7.532 (± 0.260) | -6.263 (± 0.165) | -7.869 (± 0.650) | -8.040 (± 0.337) | -7.816 (± 0.133) |
| HierVAE (Jin et al., 2020a) | -9.487 (± 0.278) | -6.812 (± 0.274) | -8.081 (± 0.252) | -8.978 (± 0.525) | -8.285 (± 0.370) |
| RationaleRL (Jin et al., 2020b) | -10.663 (± 0.086) | -8.129 (± 0.048) | -9.005 (± 0.155) | *No hit found* | -9.398 (± 0.076) |
| FREED (Yang et al., 2021) | -10.579 (± 0.104) | -8.378 (± 0.044) | -10.714 (± 0.183) | -10.561 (± 0.080) | -9.735 (± 0.022) |
| PS-VAE (Kong et al., 2022) | -9.978 (± 0.091) | -8.028 (± 0.050) | -9.887 (± 0.115) | -9.637 (± 0.049) | -9.464 (± 0.129) |
| MOOD (Lee et al., 2023) | -10.865 (± 0.113) | -8.160 (± 0.071) | -11.145 (± 0.042) | -11.063 (± 0.034) | -10.147 (± 0.060) |
| GDSS (Jo et al., 2022) | -9.967 (± 0.028) | -7.775 (± 0.039) | -9.459 (± 0.101) | -9.224 (± 0.068) | -8.926 (± 0.089) |
| RetMol (Wang et al., 2023) | -8.590 (± 0.475) | -5.448 (± 0.688) | -6.980 (± 0.740) | -8.811 (± 0.574) | -7.133 (± 0.242) |
| GEAM (Lee et al., 2024) | -12.536 (± 0.115) | -9.611 (± 0.019) | -12.385 (± 0.059) | -12.165 (± 0.092) | -11.822 (± 0.035) |
| **BindMol** (ours) | **-12.910** (± 0.077) | **-10.121** (± 0.035) | **-12.791** (± 0.102) | **-12.507** (± 0.087) | **-11.849** (± 0.019) |

Table 3: **#Circles of generated hit molecules.** The #Circles threshold is set to 0.75. The results are the means and the standard deviations of 3 runs. The best results are highlighted in bold.

| Method | Target protein | | | | |
|---|---|---|---|---|---|
| | parp1 | fa7 | 5ht1b | braf | jak2 |
| REINVENT (Olivecrona et al., 2017) | 44.2 (± 15.5) | 23.2 (± 6.6) | 138.8 (± 19.4) | 18.0 (± 2.1) | 59.6 (± 8.1) |
| MORLD (Jeon & Kim, 2020) | 1.4 (± 1.5) | 0.2 (± 0.4) | 22.2 (± 16.1) | 1.4 (± 1.2) | 6.6 (± 3.7) |
| HierVAE (Jin et al., 2020a) | 4.8 (± 1.6) | 0.8 (± 0.7) | 5.8 (± 1.0) | 3.6 (± 1.4) | 4.8 (± 0.7) |
| RationaleRL (Jin et al., 2020b) | 61.3 (± 1.2) | 2.0 (± 0.0) | **312.7** (± 6.3) | 1.0 (± 0.0) | **199.3** (± 7.1) |
| FREED (Yang et al., 2021) | 34.8 (± 4.9) | 21.2 (± 4.0) | 88.2 (± 13.4) | 34.4 (± 8.2) | 59.6 (± 8.2) |
| PS-VAE (Kong et al., 2022) | 38.0 (± 6.4) | 18.0 (± 5.9) | 180.7 (± 11.6) | 16.0 (± 0.8) | 83.7 (± 11.9) |
| MOOD (Lee et al., 2023) | 86.4 (± 11.2) | 19.2 (± 4.0) | 144.4 (± 15.1) | 50.8 (± 3.8) | 81.8 (± 5.7) |
| GEAM (Lee et al., 2024) | 123.0 (± 7.8) | 79.0 (± 9.2) | 144.3 (± 8.6) | 84.7 (± 8.6) | 118.3 (± 0.9) |
| **BindMol** (ours) | **143.1** (± 5.2) | **93.7** (± 4.5) | 150.4 (± 7.1) | **107.2** (± 4.8) | 134.0 (± 1.5) |

## 5 EXPERIMENTS

To comprehensively validate the effectiveness of our proposed **BindMol**, we conduct comprehensive experiments on two important scenarios in drug discovery: (1) *binding affinity optimization* (Section 5.1), and (2) *similarity-based rediscovery* (Section 5.2), both of which can be formulated as multi-objective optimization tasks. We further demonstrate the effectiveness of our model design through ablation experiments and case studies in Section 5.3.

**Experimental Settings** We adhere to the task settings proposed by Lee et al. (2024) for fair comparison. The policy model performs random sampling for the first 4,000 steps to collect experience, and parameter updates after 3,000 steps to ensure a sufficient replay buffer. For evaluation, we follow established metrics including *Novel hit raio* and *Novel top 5% Docking Score* for hit discovery (Yang et al., 2021), area under the curve (AUC) of top-K value for rediscovery (Gao et al., 2022), and chemical space coverage (#Circles) (Xie et al., 2023) for diversity measure. Please refer to Section B for a comprehensive description of our exmperimental settings and evaluation protocals.

## 5.1 Optimization of Binding Affinity

For binding affinity optimization, we aim to design novel, drug-like, and synthesizable binders to the specific target. Therefore, we validate `BindMol` on five protein targets, **parp1**, **fa7**, **5ht1b**, **braf**, and **jak2**, under the quantitative estimate of drug-likeness (QED) (Bickerton et al., 2012), synthetic accessibility (SA) (Ertl & Schuffenhauer, 2009), and novelty constraints following Lee et al. (2023).

**Baselines.** To comprehensively evaluate the effectiveness of our approach, we select 11 competitive molecular generation models as baselines: REINVENT (Olivecrona et al., 2017), Graph GA (Jensen, 2019), MORLD (Jeon & Kim, 2020), HierVAE (Jin et al., 2020a), RationaleRL (Jin et al., 2020b), FREED (Yang et al., 2021), PS-VAE (Kong et al., 2022), MOOD (Lee et al., 2023), GDSS (Jo et al., 2022), RetMol (Wang et al., 2023), and GEAM (Lee et al., 2024). These baselines span both SMILES- and graph-based modalities, and encompass diverse training paradigms including distribution learning, genetic algorithms, and reinforcement learning.

**Performance comparison.** Empirical results are summarized in Tables 1 to 3. Our systematic study suggests the following trends: *(i) `BindMol` achieves state-of-the-art performance across all targets*, highlighting its ability to generate novel, drug-like, synthesizable compounds with strong binding affinity. In particular, for challenging targets such as *fa7*, our method attains a novel hit ratio above 20%, surpassing GEAM (17%) and far outperforming most competing methods, many of which approach zero. *(ii) Fragment-based reinforcement learning methods are especially effective at exploring high-reward regions.* FREED, MOOD, GEAM, and our `BindMol` all adopt fragment-based RL strategies and consistently produce lower docking scores. This advantage arises from a well-defined action space: fragment-level operations are limited yet high-quality, enabling more efficient exploration than token-based SMILES generation. *(iii) `BindMol` achieves higher #Circle while maintaining a strong hit ratio*, indicating that the generated molecules are not only effective but also diverse. This reflects the benefit of explorative preferences, which encourage RL to explore a wider range of action sequences and thereby cover chemical space more comprehensively. By contrast, RationaleRL achieves relatively high diversity on *5ht1b* and *jak2*, but its overly aggressive exploration leads to very low hit discovery rates.

**Multi-residue exploration.** To evaluate the effectiveness of `BindMol` in exploring binding modes, we employ the **Hypervolume** metric, which measures the volume of the space dominated by the Pareto front of solutions and bounded by the preference point. Since multi-objective learning for residues often involves a large number of objectives (typically 5–30), most existing multi-objective algorithms become computationally infeasible. Therefore, we compare our approach with Lee et al. (2024), the best-performing method on binding tasks. The results are shown in Figure 3. It can be observed that our method achieves a more thorough exploration of the Pareto front across all five targets, demonstrating that `BindMol` can generate molecules with more diverse binding modes.

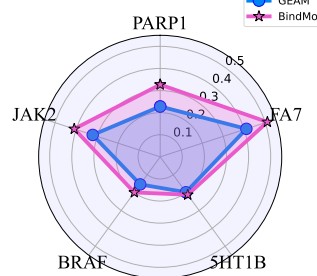

Figure 3: The Hypervolume comparison of `BindMol`.

## 5.2 Optimization of Multi-property Objectives in GuacaMol

For similarity-based rediscovery, our goal is to generate molecules similar to the target compound and fulfill multiple required property profiles. Following Gao et al. (2022), we select the multi-property optimization tasks from the GuacaMol benchmarks (Brown et al., 2019) for capability evaluation. Notably, unlike prior studies that scalarize multiple objectives into a single target, we treat them as a genuine multi-objective problem to better match our algorithm. To ensure fairness, we use each individual objective score prior to scalarization as a single-objective reward.

**Baselines.** We select 12 molecular generation models with strong performance on the GuacaMol benchmark as baselines: REINVENT (Olivecrona et al., 2017), MolDQN (Zhou et al., 2019), Graph GA (Jensen, 2019), Graph MCTS (Jensen, 2019), DoG-Gen (Bradshaw et al., 2020), GPBO (Tripp et al., 2021), STONED (Nigam et al., 2021), SynNet (Gao et al., 2021), DST (Fu et al., 2021), GEAM (Lee et al., 2024), Augmented Memory (Guo & Schwaller, 2024), and MOLLEO (MoleculeSTM) (Wang et al., 2025). These methods also span various molecular modalities and training strategies.

**Performance comparison.** Empirical results for MPO tasks are presented in Table 4 and Table 5. It can be observed that `BindMol` significantly outperforms all baseline models across all MPO

Table 4: **GuacaMol MPO AUC Top-100 results.** The results are the means of 3 runs. We highlight the best-
and the second-performing results in **boldface** and underlined, respectively.

| Method | Benchmark | | | | | | | Average |
|---|---|---|---|---|---|---|---|---|
| | Amlodipine | Fexofenadine | Osimertinib | Perindopril | Ranolazine | Sitagliptin | Zaleplon | |
| REINVENT (Olivecrona et al., 2017) | 0.608 | 0.752 | 0.806 | 0.511 | 0.719 | 0.006 | 0.325 | 0.532 |
| MolDQN (Zhou et al., 2019) | 0.230 | 0.431 | 0.636 | 0.125 | 0.018 | 0.000 | 0.002 | 0.206 |
| Graph GA (Jensen, 2019) | 0.622 | 0.731 | 0.799 | 0.503 | 0.670 | 0.330 | 0.305 | 0.566 |
| Graph MCTS (Jensen, 2019) | 0.385 | 0.522 | 0.655 | 0.219 | 0.694 | 0.117 | 0.165 | 0.394 |
| DoG-Gen (Bradshaw et al., 2020) | 0.489 | 0.640 | 0.706 | 0.422 | 0.601 | 0.015 | 0.073 | 0.421 |
| GPBO (Tripp et al., 2021) | 0.538 | 0.685 | 0.750 | 0.460 | 0.694 | 0.117 | 0.165 | 0.487 |
| STONED (Nigam et al., 2021) | 0.593 | 0.777 | 0.799 | 0.472 | 0.738 | 0.351 | 0.307 | 0.577 |
| SynNet (Gao et al., 2021) | 0.533 | 0.720 | 0.759 | 0.512 | 0.690 | 0.007 | 0.223 | 0.492 |
| DST (Fu et al., 2021) | 0.469 | 0.690 | 0.742 | 0.425 | 0.579 | 0.017 | 0.089 | 0.430 |
| GEAM (Lee et al., 2024) | 0.627 | 0.786 | 0.817 | 0.514 | 0.714 | 0.408 | 0.386 | 0.607 |
| Augmented Memory (Guo & Schwaller, 2024) | 0.607 | 0.657 | 0.818 | 0.518 | 0.725 | 0.398 | 0.355 | 0.582 |
| MOLLEO (MoleculeSTM) (Wang et al., 2025) | 0.617 | 0.765 | 0.799 | 0.516 | 0.675 | 0.415 | 0.392 | 0.597 |
| **BindMol** (Ours) | **0.663** | **0.839** | **0.838** | **0.525** | **0.742** | **0.443** | **0.412** | **0.637** |

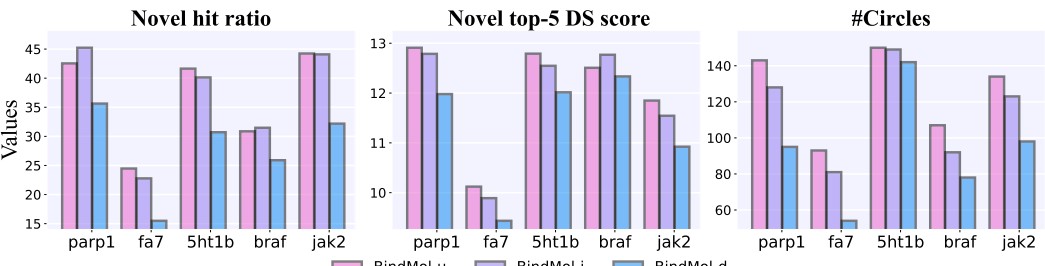

Figure 4: Performance of different ablation variants of `BindMol`. DS denotes Docking Score.

tasks, which further demonstrates the effectiveness of the proposed Envelope SAC algorithm. On the
two rediscovery tasks of *Amlodipine* and *Fexofenadine*, it achieves relative improvements of 6.7%
and 5.7% over the second-best models, respectively. Moreover, on the most challenging *Sitagliptin*
rediscovery task, `BindMol` attains a record-breaking score of 0.443. It is worth noting that MOLLEO
(MoleculeSTM), a framework that applies genetic algorithms over large-scale molecular corpora with
language models, is still outperformed by our method on rediscovery tasks. We exclude other variants
of MOLLEO (e.g., those using GPT-4 as the base LLM) from comparison due to their excessive
unfair advantage in this setting.

## 5.3 ABLATION STUDIES

We conduct extensive ablation studies on several variants of our framework. To identify key residues,
we define the target set either as the union or the intersection of residues obtained from different
complexes, leading to the variants `BindMol`-*u* and `BindMol`-*i*, respectively. We also ablate the
residue-level reward by using only the docking score for each target as the final reward, resulting in
the variant `BindMol`-*d*. As shown in Figure 4, both the union and intersection strategies achieve
comparable performance on novel hit ratio and novel top-5% docking score, and both clearly
outperform the variant without residue-level rewards. This indicates that reformulating traditional
docking score optimization into multi-residue optimization substantially improves hit discovery
efficiency. Furthermore, `BindMol`-*i* yields noticeably lower #Circle compared to `BindMol`-*u*,
suggesting that broader residue selection facilitates more comprehensive chemical space coverage,
while `BindMol`-*d* exhibits the weakest coverage due to the absence of residue-level optimization.

## 5.4 CASE STUDIES

To provide an intuitive understanding of the binding modes exploration of `BindMol`, we cluster the
generated molecules according to the number of protein–ligand interactions across different targets
using $k$-means ($k = 2$). For each cluster, we select and visualize representative samples to examine
their interaction patterns. Specifically, we perform clustering analysis on the *parp1* and *jak2* targets,
resulting in four protein–ligand complexes for visualization, as shown in Figure 5.

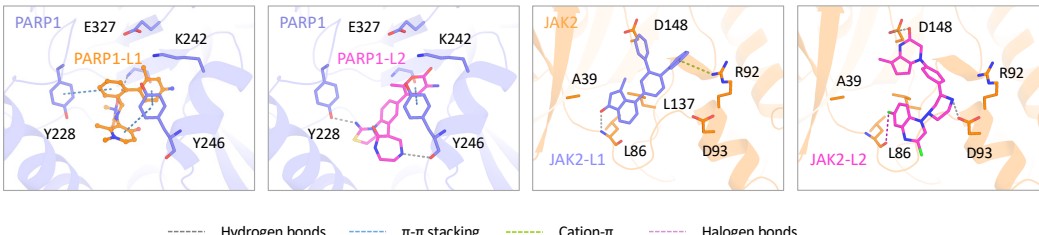

Figure 5: Visualization of representative binding modes on two protein comlexes: *parp1* and *jak2*.

In the *parp1* complexes, the two ligands exhibit distinct binding modes: PARP1-L1, with a polyaromatic scaffold, is dominated by aromatic stacking with Y228 and Y246, whereas PARP1-L2, featuring a heteroaromatic scaffold, relies more heavily on hydrogen bonding with diminished stacking. In the *jak2* complexes, the differences are also evident. JAK2-L1 adopts a stacking- and cation–π–driven mode, while JAK2-L2 introduces heteroatoms and halogen substitution that favor additional hydrogen and halogen bonds, reducing dependence on R92.

## 6 CONCLUSION

In this work, we introduce a paradigm shift in goal-directed drug discovery by reformulating the task as a multi-objective exploration problem, thereby moving beyond the limitations of monolithic scalar rewards. We present **BindMol**, a novel framework that integrates a fragment-based generator with a customized multi-objective reinforcement learning algorithm, *Envelope SAC*. By decomposing the reward function into residue-specific interaction objectives, BindMol effectively learns to explore a diverse landscape of binding modes. Our comprehensive experiments demonstrate that this approach establishes new state-of-the-art performance on both challenging binding affinity optimization targets and standard GuacaMol benchmarks, consistently discovering novel, high-affinity molecules with greater interaction diversity. By enabling a more intelligent and controllable search of chemical space, our work provides a powerful and versatile paradigm for designing compounds with specific and desirable pharmacological profiles, significantly advancing the toolkit for AI-based drug discovery.

**Limitations and future works.** While **BindMol** demonstrates strong performance in exploring diverse binding modes, our current reward formulation relies on simplified interaction counts and static docking poses. First, treating all interaction types equally—without weighting them by energetic contribution—prioritizes interaction diversity over fine-grained affinity modeling. Assigning biochemically accurate weights is non-trivial and target-dependent, yet future iterations could leverage learnable or physics-based weighting schemes to better capture residue-specific importance. Second, like most structure-based generative models, our reliance on rapid static docking (e.g., QVina) may occasionally reward suboptimal or physically strained poses. We aim to address it in future work by incorporating explicit structural constraints and rigorous energy minimization steps to filter out spurious interactions and ensure physically plausible binding configurations.

## 7 REPRODUCIBILITY STATEMENT

We have taken several steps to facilitate the reproducibility of our results. Section 5 provides a summary of the experimental settings, while Appendix B gives detailed descriptions of the computing infrastructures, implementation details, and evaluation protocols. In addition, we include the source code as a compressed archive in the supplementary materials to further support reproducibility.

## 8 THE USE OF LARGE LANGUAGE MODELS (LLMS)

Large language models were used solely for linguistic refinement and polishment of the manuscript. All scientific ideas, methodological contributions, and experimental results are entirely conceived, implemented, and validated by the authors.

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

APPENDIX

# A DYNAMIC FRAGMENT VOCABULARY

Our policy model for molecular generation relies on a dynamic fragment vocabulary, following (Lee et al., 2024). This design avoids the limitations of a fixed set of building blocks by periodically identifying new, high-scoring fragments from newly generated molecules. The core of this process is a goal-aware fragment extraction model, which we detail below.

## A.1 GOAL-AWARE FRAGMENT EXTRACTION.

Given a dataset of molecules and their associated properties, the goal is to identify fragments that are most predictive of a desired property. It is framed as an information bottleneck problem (Wu et al., 2020), where we aim to find a compressed representation of a molecule that retains maximal information about the property while discarding irrelevant structural data.

First, for a given molecule graph $G = (\mathcal{V}, \mathcal{E})$, we adopt a message passing neural network (MPNN) (Gilmer et al., 2017) to compute the node embeddings $\{\mathbf{h}_v \in \mathbb{R}^d \mid v \in \mathcal{V}\}$. The molecule is then decomposed into a set of fragments $\{F_j = (V_j, E_j)\}$ using BRICS algorithm (Degen et al., 2008). The representation for each fragment, $\mathbf{e}_j$, is obtained by averaging the embeddings of its constituent nodes:

$$\mathbf{e}_j = \frac{1}{|V_j|} \sum_{v \in V_j} \mathbf{h}_v. \tag{13}$$

Next, to quantify the relevance of each fragment to the target property, a Multi-Layer Perceptron (MLP) computes an importance weight $w_j \in [0, 1]$ from its embedding, i.e., $w_j = \text{MLP}(\mathbf{e}_j)$. This weight controls an information bottleneck, where irrelevant fragments are masked with noise following (Yu et al., 2022). The perturbed fragment embedding $\tilde{\mathbf{e}}_j$ is formulated as:

$$\tilde{\mathbf{e}}_j = w_j \mathbf{e}_j + (1 - w_j) \bar{\boldsymbol{\mu}} + \boldsymbol{\eta}, \quad \text{where} \quad \boldsymbol{\eta} \sim \mathcal{N}(\mathbf{0}, (1 - w_j) \bar{\boldsymbol{\Sigma}}). \tag{14}$$

Here, $\bar{\boldsymbol{\mu}}$ and $\bar{\boldsymbol{\Sigma}}$ are the empirical mean and diagonal covariance of all fragment embeddings. The entire model is trained by minimizing an objective derived from the information bottleneck principle, which balances a property prediction loss with a regularization term that encourages compression:

$$\mathcal{L} = \mathbb{E}_{G, Y \sim \mathcal{D}} \left[ \mathcal{L}_{\text{pred}}(Y, Z) + \beta \, \text{KL}[p(Z|G) \| u(Z)] \right], \tag{15}$$

where $Z = \{\tilde{\mathbf{e}}_j\}$ represents the collection of perturbed fragment embeddings, $\mathcal{L}_{\text{pred}}$ is the prediction loss (e.g., cross-entropy), and the KL-divergence term penalizes information retained from the original graph $G$.

## A.2 FRAGMENT SCORING AND VOCABULARY UPDATE

After training, the model is used to score all candidate fragments. The score for a fragment $F$ reflects its average contribution to achieving a high property value across all molecules it appears in. It is calculated as:

$$\text{Score}(F) = \mathbb{E}_{(M,y) \in \mathcal{D}_F} \left[ \frac{w(F, M)}{\sqrt{|V_F|}} \cdot y \right], \tag{16}$$

where $\mathcal{D}_F$ is the set of molecules containing fragment $F$, $w(F, M)$ is the learned importance of $F$ within molecule $M$, $|V_F|$ is the number of heavy atoms in the fragment (a size normalization term), and $y$ is the molecule's property value. Fragments with the highest scores are selected to populate the vocabulary $\mathcal{S}$, which is then used for the generative process and cyclically updated.

# B EXPERIMENTAL SETTINGS

## B.1 COMPUTING INFRASTRUCTURES

**Software infrastructures.** All of the experiments are implemented in Python 3.10, with the following supporting libraries: PyTorch 2.1.0 (Paszke et al., 2019), PyG 2.0.4 (Fey & Lenssen, 2019), DGL 2.4.0 (Wang, 2019), Gym 0.21.0 (Brockman et al., 2016), Open Babel 3.1.1 (O'Boyle et al., 2011), PLIP 2.3.1 (Salentin et al., 2015), and RDKit 2022.03.1 Landrum et al. (2016).

**Hardware infrastructures.** We conduct all experiments on a computer server with 8 NVIDIA A800-SXM4-40GB GPUs (with 24GB memory each) and 2 Intel Xeon Gold 6348 CPUs (with 56 cores and 112 threads).

## B.2 IMPLEMENTATION DETAILS

Following Yang et al. (2021); Gao et al. (2022); Lee et al. (2024), we base our experimental setup on the ZINC250k dataset (Irwin et al., 2012), using the standard split for training and testing. For our generative model,the training process begins with an initial fragment vocabulary of 300 fragments, which is dynamically expanded to a maximum size of 1000 by adding up to 50 new fragments per cycle. Any fragments that cause RDKit sanitization errors are filtered out.

The fragment-based generator employs Envelope SAC for training, where episode starts from a benzene ring scaffold with three attachment points (*ortho-*, *meta-*, *para-*). An episode terminates if the heavy atom count of molecule exceeds 40. To ensure sufficient exploration, the policy model samples molecules randomly for the first 4,000 steps to populate the replay buffer, while parameter updates commence after 3,000 steps. For fair evaluation, molecules generated during this initial exploration phase are included in the final results. The fragment modification module utilizes a genetic algorithm with a population size of 100 and a mutation rate of 0.1.

All neural networks use a hidden dimension of 64 and are trained with a batch size of 256. Both the value and policy networks are optimized using the Adam optimizer (Kingma & Ba, 2014) with a learning rate of $1 \times 10^{-4}$. For exploration in our multi-objective setting, we sample 5 preference vectors per batch. Moreover, we observe that as the number of objectives increases, the training benefits from a higher ratio of off-policy updates (i.e., utilizing more samples from the replay buffer relative to on-policy samples). We hypothesize that in high-dimensional objective spaces, the model requires more frequent re-visitation of past action episodes to effectively learn the complex trade-offs and stabilize convergence.

## B.3 EVALUATION PROTOCOLS

We adopt several well-defined metrics to evaluate our framework:

- For binding affinity optimization, we evaluate the generated molecules following Yang et al. (2021). A molecule is considered a *hit* if it achieves a docking score lower than the median of experimentally known actives, while also satisfying QED > 0.5 and SA < 5. To assess affinity optimization, we report the ***Novel top 5% Docking Score*** (kcal/mol), which is the average docking score of the best-performing 5% unique and novel hits. We also measure the ***Novel hit ratio***, defined as the fraction of hits that are novel.

- For rediscovery tasks, we report the area under the curve (AUC) of top-K average property value versus the number of oracle calls (***AUC top-K***) following Gao et al. (2022) for fair comparison.

- We further report ***#Circles*** (Xie et al., 2023) for both of the tasks, which quantifies the coverage of chemical space by the generated hits.

## C ADDTIONAL EMPIRICAL RESULTS

Table 5: **GuacaMol MPO #Circles results.** The results are the means of 3 runs. The best and second-best results are highlighted in bold and underlined, respectively.

| Method | Benchmark | | | | | | |
|---|---|---|---|---|---|---|---|
| | Amlodipine | Fexofenadine | Osimertinib | Perindopril | Ranolazine | Sitagliptin | Zaleplon |
| REINVENT (Olivecrona et al., 2017) | 303.7 | 343.3 | 452.3 | 318.3 | 253.3 | **398.3** | 275.3 |
| Graph GA (Jensen, 2019) | 258.7 | 333.3 | 270.3 | 278.7 | 364.7 | 306.3 | 272.7 |
| STONED (Nigam et al., 2021) | 303.7 | 330.3 | 301.3 | 301.0 | 316.7 | 326.3 | 280.3 |
| GEAM-static (Lee et al., 2024) | 412.0 | 397.7 | 315.3 | 318.0 | 256.7 | 233.0 | 267.0 |
| GEAM (Lee et al., 2024) | 424.0 | 502.0 | 435.0 | **377.3** | 295.3 | 257.0 | 336.0 |
| **BindMol** (Ours) | **486.2** | **538.0** | **471.6** | 375.5 | **402.0** | 353.0 | **357.3** |

## C.1 SENSITIVITY OF PREFERENCE SAMPLING SIZE

As mentioned in Section 4.2, the maximization over $\omega'$ requires approximation. It is sampled by first drawing i.i.d. standard normal vectors from $\mathcal{N}(\mathbf{0}, \mathbf{I})$, applying an element-wise absolute value to ensure non-negativity, and then performing $\ell_1$ normalization. This ensures that the resulting vectors are uniformly distributed on the probability simplex. These weights are then replicated across the batch dimension to enable efficient preference-conditioned training. However, since replicating preference vectors for a batch of PyG graphs increases memory usage, very dense sampling would significantly increase the GPU footprint. In practice, we therefore sample $\omega$ five times, which provides a good balance between computational efficiency and performance. To further demonstrate the effect of sampling size $k$, we have conducted a sensitivity analysis and the results are shown in Table 6.

Table 6: Sensitivity analysis of preference sampling size $k$.

| Novel hit ratio (%) | parp1 | fa7 | 5ht1b | braf | jak2 |
|---|---|---|---|---|---|
| $k = 1$ | 35.025 | 18.277 | 36.272 | 19.333 | 39.047 |
| $k = 3$ | 38.903 | 21.667 | 39.622 | 25.887 | 42.887 |
| $k = 5$ (**BindMol**) | 42.538 | 24.471 | 41.639 | 30.867 | 44.238 |
| $k = 10$ | 43.622 | 24.891 | 42.277 | 31.047 | 45.219 |
| $k = 20$ | 45.667 | 25.891 | 44.239 | 33.127 | 46.975 |

It can be observed that with samping size increase, the performance of novel hit ratio on three targets steadily increase. However, due to the memory burden, we set sampling size to $k = 5$ as the final configuration to achieve the trade-off between performance and efficiency.

## C.2 MOLECULAR WEIGHT DISTRIBUTION ANALYSIS

To investigate whether our per-residue interaction rewards unintentionally incentivize the generation of larger molecules, we conducted a comparative analysis of the molecular weight (MW) distributions produced by **BindMol** and several representative baselines. For each method, we sample an equal number of generated molecules for the target jak2 and compute their MW values using RDkit. As an external reference, we also included the MW of experimentally validated ligand extracted from the PDB (PDB id: 7F7W). The results are shown in Figure 6.

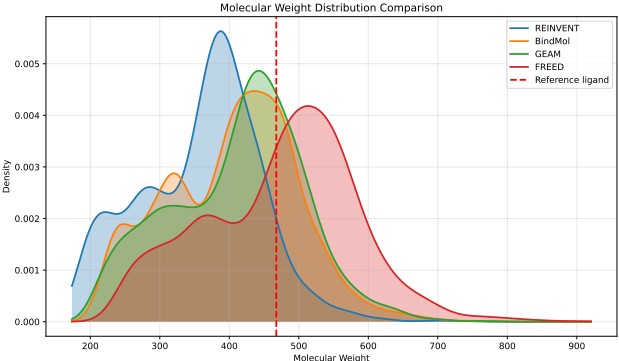

Figure 6: The comparison of molecular weight of generated molecules on jak2 target using different competitive generative approaches.

As shown in Figure 6, the MW distribution of **BindMol** closely aligns with those of the baseline methods and is broadly consistent with the distribution observed in PDB ligands. These results indicate that our per-residue interaction rewards do not systematically drive the model toward generating heavier molecules. The similarity between generated molecules and reference ligands further suggests that we operate within a chemically reasonable MW range that is typical for bioactive small molecules.

# D  VISUALIZATION OF GENERATED MOLECULES

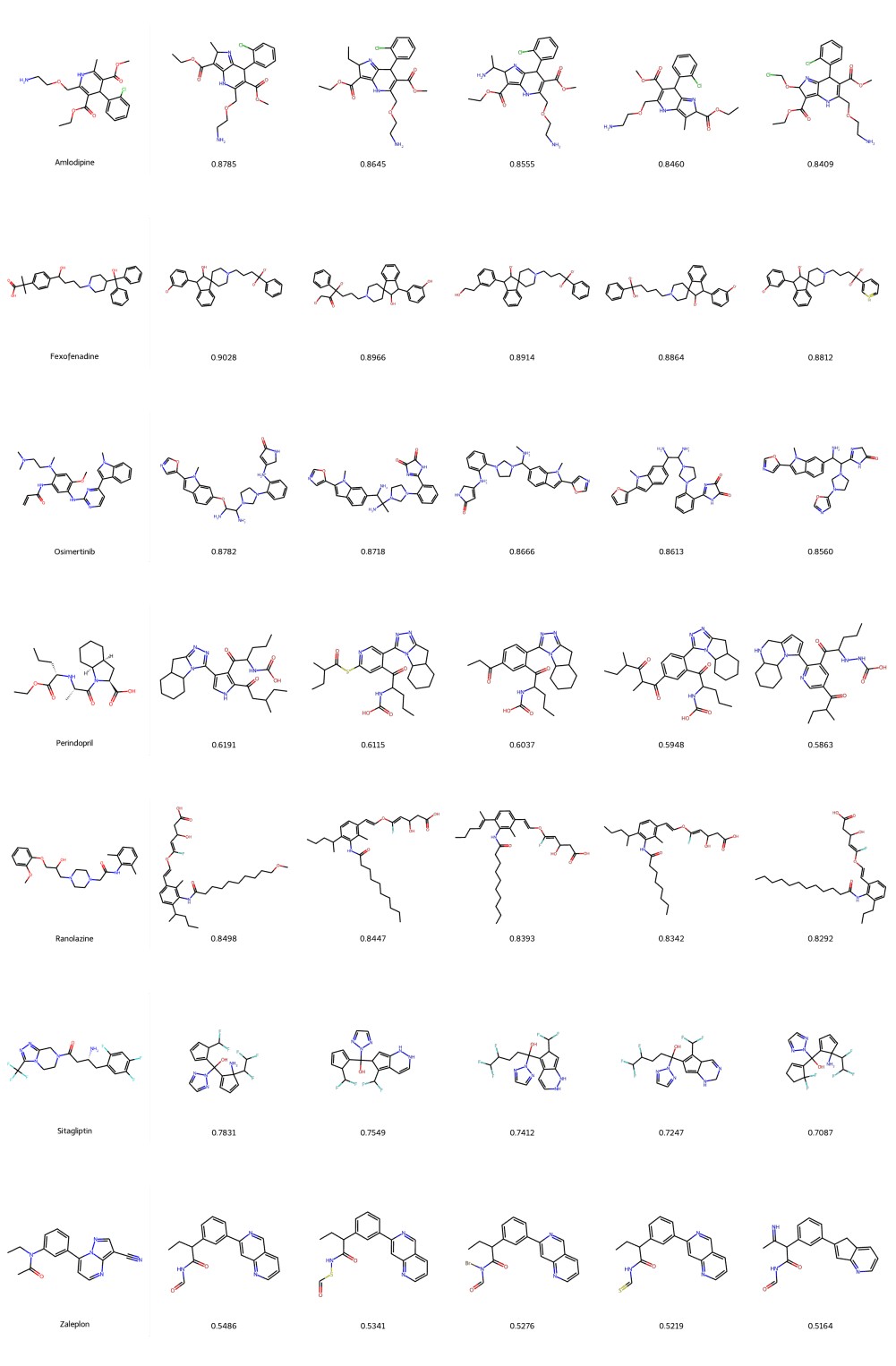

Figure 7: The rediscovery molecules of the GuacaMol MPO tasks and the examples of the generated top-5 molecules. The scores are provided at the bottom of each generated molecule.

