# OpenReview forum: "Binding Mode Matters: Residue-Guided Drug Discovery via Explorative Preferences"
_ICLR.cc/2026/Conference — Submitted to ICLR 2026_

### Official Review · Reviewer_dzp3 · 2025-10-28

**Soundness:** 2
**Presentation:** 2
**Contribution:** 1
**Rating:** 4
**Confidence:** 4

**Summary:**

Through this paper, the authors propose BindMol, a fragment-based molecule generation framework using a multi-objective RL algorithm. Specifically, the authors newly propose incorporating explorative preferences during training. The experiments show that BindMol can discover molecules with good binding profiles.

**Strengths:**

- The authors provided the codebase.
- The proposed formulation that views target-based drug discovery as a multi-objective optimization problem, with each objective corresponding to a key residue within the binding pocket, is interesting and reasonable.
- The proposed BindMol shows good performance on various benchmarks.

**Weaknesses:**

Weaknesses
I will combine the *Weaknesses* section and the *Questions* section. My concerns are as follows:
- The main weakness of this paper is its weak novelty. As the authors mentioned in lines 56~58, the BindMol framework consists of three key components: action space, reward function, and the RL algorithm. For the action space, BindMol adopts FREED [1]. Section 4.1 actually should actually be placed in Preliminaries section and is not an invention of this work. BindMol's reward function design (Section 4.3) is a heuristic that relies on the PLIP tool and docking scores, and it cannot be considered a significant contribution from an ML perspective. The only main contribution of this work is Envelope SAC, a multi-objective RL algorithm that defines a preference-aware Bellman operator and a vectorized Q-function (Section 4.2). However, the central idea is to integrate the envelope-based update mechanism is very similar to MORL [2]. Overall, I am not convinced that this work provides a new approach compared to previous methods in the domain.
- In the GuacaMol MPO experiment (Section 5.2, Table 4), SOTA molecular optimization baselines such as GenMol [3] and Genetic GFN [4] are missing. Comparisons with these baselines are necessary for the results to be considered meaningful.

---

**References:**

[1] Yang et al., Hit and lead discovery with explorative rl and fragment-based molecule generation, NeurIPS, 2021.

[2] Yang et al., A generalized algorithm for multi-objective reinforcement learning and policy adaptation, NeurIPS, 2019.

[3] Lee et al., GenMol: A Drug Discovery Generalist with Discrete Diffusion, ICML 2025.

[4] Kim et al., Genetic-guided GFlowNets for Sample Efficient Molecular Optimization, NeurIPS 2024.

**Questions:**

Please see the *Weaknesses* section for my main concerns.

---

> ### Author Response · Authors · 2025-11-21
> **Response to Reviewer dzp3**
>
> > **W1: The main weakness of this paper is its weak novelty.**
>
> We thank the reviewer for the detailed assessment. We recognize that BindMol builds upon established foundations—such as fragment-based generation and envelope Q-learning concepts. However, we respectfully disagree with the conclusion that the novelty is limited or merely incremental. We believe the contribution of BindMol lies in the **non-trivial integration** of these components to solve a previously unaddressed biological challenge. We clarify our specific contributions below:
>
> **(1) Algorithmic Novelty: Envelope SAC is not just "MORL + SAC".**
> While the reviewer correctly identifies the connection to the envelope Q-learning in MORL [2], it is crucial to highlight that the original envelope operator was designed strictly for **value-based methods (e.g., DQN)**. Extending this to an actor-critic framework (SAC) is practically non-trivial. Unlike value-based methods, Actor-Critic requires the simultaneous optimization of a *preference-conditioned Policy* (Actor) and a *multi-objective Critic*. Ensuring the consistency between the actor’s policy update and the critic’s preference alignment presents unique optimization challenges that do not exist in pure value-based settings. Our proposed Envelope SAC is the first to solve this alignment and integration problem, offering a principled solution for continuous control in chemical space.
>
> **(2) Conceptual Novelty: A Paradigm Shift in Reward Formulation.**
> We respectfully argue that our reward design (Section 4.3) should not be viewed merely as a heuristic use of PLIP.
> Existing methods treat binding strictly as a scalar optimization problem (collapsing all interactions into a single score), which inherently ignores the structural diversity of ligand-target interactions. By formulating these interactions as a **vectorized objective**, we are the first to treat **binding-mode diversity as a first-class modeling objective**. This is a conceptual leap from "optimizing a score" to "exploring biological mechanisms," which we believe constitutes a significant contribution to the machine learning for drug discovery community.
>
> For a more detailed elaboration on these points, we kindly invite the reviewer to refer to our `General Response`. We sincerely hope these clarifications address your concerns regarding novelty and warrant a re-evaluation of our work's contribution.
>
>
>
> > **W2: In the GuacaMol MPO experiment (Section 5.2, Table 4), SOTA molecular optimization baselines such as GenMol [3] and Genetic GFN [4] are missing. Comparisons with these baselines are necessary for the results to be considered meaningful.**
>
> We thank the reviewer for bringing these relevant works to our attention. While we acknowledge their strong performance on GuacaMol benchmarks, we respectfully emphasize a **fundamental distinction in the experimental setting**. Both GenMol [3] and Genetic GFN [4] benefit significantly from **large-scale pretraining** on massive external datasets (e.g., UniChem and ZINC). In contrast, BindMol is designed as a **train-from-scratch** framework; it optimizes molecules solely through interaction with the oracle and an initial fragment library, without leveraging the massive data-driven priors inherent in pretraining.
>
> Therefore, to ensure a fair comparison regarding data efficiency and prior knowledge, we restricted our baselines to methods that operate under similar "cold-start" conditions. We believe that directly comparing our method against pretrained models would conflate the algorithmic contribution with the advantages derived from external large-scale data.

---

> ### Author Response · Authors · 2025-11-28
> **Respectful Inquiry Before Discussion Deadline**
>
> Dear Reviewer dzp3,
>
> Hope this message finds you well. We wanted to gently remind you that the deadline for the discussion phase is approaching, and we would greatly appreciate it if you could take a moment to review our responses.
>
> Your feedback is very valuable to us, and we wanted to ensure all your concerns have been addressed satisfactorily. If there are any additional concerns or points of clarification, we are more than happy to address them.
>
> Thank you for your time and consideration. We eagerly await your further guidance with utmost respect.
>
> Best wishes,
>
> Authors

---

### Official Review · Reviewer_Utui · 2025-10-31

**Soundness:** 2
**Presentation:** 2
**Contribution:** 2
**Rating:** 4
**Confidence:** 3

**Summary:**

This paper proposes BindMol, a fragment based molecular generation framework for structure guided drug design that treats binding as a multi objective reinforcement learning problem. Instead of optimizing a single docking score, the method defines residue level rewards that capture interactions with individual protein residues and trains a policy using an Envelope Soft Actor Critic algorithm to explore diverse binding preferences. The approach constructs molecules by sequentially attaching fragments and leverages a dynamic vocabulary to expand chemical space during training. Experiments on five protein targets and multi property GuacaMol benchmarks show improved hit rates, docking scores, and chemical diversity compared to strong baselines, indicating that residue guided rewards can promote diverse binding modes and high affinity candidates

**Strengths:**

- The paper introduces a biologically motivated multi objective formulation for structure based molecular generation, where residue level interaction signals replace a single scalar docking score, providing a more interpretable and controllable reward structure.
- The Envelope SAC algorithm with preference based exploration is a technically novel component that extends entropy regularized RL to vector valued rewards and empirically improves Pareto frontier coverage in binding mode space.
- Experimental evaluation is broad and competitive, covering five protein binding tasks and seven GuacaMol multi property benchmarks, and demonstrating consistent state of the art performance in affinity, novelty, and diversity metrics.

**Weaknesses:**

- The connection between the scalar reward definition in Section 4.3 and the multi-objective formulation in Section 4.2 remains ambiguous. It is not clearly explained how the final scalar reward integrates into the Envelope SAC optimization process.
- The residue-based reward design merely counts the number of interactions without reflecting residue-specific importance or interaction strength. This simplification may limit the model’s ability to capture nuanced biochemical factors in binding.
- Several mathematical symbols and operators (e.g., ω*, Hα, Qθ) are introduced without full contextual definition or consistent usage, which may hinder the theoretical clarity and reproducibility of the proposed method.
- Although experiments support that multi-residue optimization improves diversity, the paper does not clearly justify why interacting with more residues should yield better ligand quality or affinity. The underlying biochemical rationale remains underexplored.
- The Envelope SAC algorithm involves solving an additional optimization (arg max over ω) during training, but the paper does not quantify the resulting computational overhead or its impact on convergence time.
- In the experimental section, it is not described how the preference vector ω is set during inference, leaving unclear how the model’s multi-objective nature is actually utilized when generating final molecules.

**Questions:**

- The reward function counts PLIP-detected interactions equally across residues. Do the authors observe cases where increasing the number of weak or geometrically marginal interactions leads to inflated reward? Have they considered weighting interaction types by estimated energetic contribution?
- The model relies on static docking poses to compute residue-level interactions. How do the authors mitigate the risk that suboptimal or strained docking poses lead to spurious interactions being rewarded?
- The dynamic fragment vocabulary is introduced as a novelty. To isolate its contribution, could the authors provide ablation results comparing a fixed-vocabulary variant with identical reward shaping?
- Multi-objective RL methods often face challenges with instability as the number of objectives grows. The paper cites settings with five to thirty residues. Could the authors report performance as a function of the number of objectives, or provide guidance on stability and hyperparameter sensitivity?

---

> ### Author Response · Authors · 2025-11-21
> **Response to Reviewer Utui (1/4)**
>
> > **W1: The connection between the scalar reward definition in Section 4.3 and the multi-objective formulation in Section 4.2 remains ambiguous. It is not clearly explained how the final scalar reward integrates into the Envelope SAC optimization process.**
>
> We thank the reviewer for the careful reading and valuable comments. We apologize for a notational error in the original manuscript that caused this confusion.
>
> The reward definitions in **Section 4.3** are formulated for **individual objective (i.e., single residue)**. The summation in the original equation should have been indexed over *interaction types* for a specific residue, rather than over the residue set. In other words, for each residue (objective), we calculate the sum of all its interaction types. Additionally, we include the docking score as a global reward component to mitigate reward sparsity in cases where residue-specific interactions are few.
>
> These per-residue rewards $\mathcal{R}(x,i)$ correspond directly to the components $\mathcal{R}_i$ of the target vector $\mathcal{R}_i$ described in `Algorithm 1`. We have corrected the notation and highlighted the relevant description in the `Section 4.3` of revised manuscript to clearly articulate the connection. We are grateful to the reviewer for pointing this out.
>
> > **W2 & Q1: The residue-based reward design merely counts the number of interactions without reflecting residue-specific importance or interaction strength. This simplification may limit the model’s ability to capture nuanced biochemical factors in binding.**
>
> Thanks for your insightful comment. We agree that different interaction types contribute differently to binding affinity. In our early exploration, we indeed considered a weighted scheme as you suggested. However, determining optimal weights is highly complex, as **the importance of specific interactions varies significantly across different targets and binding pockets**. For instance, a hydrogen bond in a buried hydrophobic pocket contributes significantly more to affinity than one on the solvent-exposed surface. Furthermore, calculating precise energetic contributions on a case-by-case basis is computationally prohibitive for RL training, which requires dense reward evaluations.
>
> Therefore, to avoid introducing excessive inductive bias and to maintain high training efficiency, we simply chose to treat all interaction types equally. Our current design prioritizes capturing the **diversity of interactions across different residues**, rather than the fine-grained energetic differences within a single residue. We greatly appreciate your suggestion and plan to explore more detailed, non-heuristic representations of interaction reward in future work. We have expanded these discussion in the `Section 6`  of revised manuscript.
>
> > **W3: Several mathematical symbols and operators are introduced without full contextual definition or consistent usage, which may hinder the theoretical clarity and reproducibility of the proposed method.**
>
> We thank the reviewer for pointing this out. In `Section 4.2`, our logical flow was to first introduce the general definition of the Envelope Q-function, and then detail the specific critic and policy updates using the instantiated model parameters. We structured it this way to avoid the clutter of complex subscripts and improve readability.
>
> We have carefully revised the manuscript to provide clearer definitions for all symbols and ensure strict consistency throughout the text. These updates have been highlighted in the revised manuscript. We hope these revisions resolve your concerns regarding theoretical clarity.

---

> ### Author Response · Authors · 2025-11-21
> **Response to Reviewer Utui (2/4)**
>
> > **W4: Although experiments support that multi-residue optimization improves diversity, the paper does not clearly justify why interacting with more residues should yield better ligand quality or affinity. The underlying biochemical rationale remains underexplored.**
>
> We thank the reviewer for raising this fundamental question regarding the biochemical rationale of our reward design. We clarify that our strategy of incentivizing interactions with multiple residues is grounded in two core principles of drug design (SBDD): **Enthalpy-driven affinity optimization** and the **Hotspot theory**.
>
> **(1) Maximizing Enthalpic Contribution via Interaction Networks.**
> From a thermodynamic perspective, binding affinity ($\Delta G$) is largely driven by the enthalpic gain ($\Delta H$) derived from specific intermolecular interactions (e.g., hydrogen bonds, salt bridges, van der Waals contacts) [1]. While ligand rigidification introduces an entropic penalty, established medicinal chemistry principles suggest that establishing a dense network of specific interactions with pocket residues is the primary strategy to overcome this penalty and achieve high-affinity binding (often referred to as "enthalpic optimization") [2]. Therefore, by encouraging the agent to interact with more residues, we are effectively guiding it to maximize the favorable enthalpic contributions that underpin high-quality binding.
>
> **(2) Anchoring to Energetic Hotspots.**
> According to the classic **Hotspot Theory** [3], a small fraction of interface residues (hotspots) often contributes the majority of the binding free energy. In our framework, the "key residues" selected for optimization are typically these functional or structural hotspots. By explicitly optimizing for interactions with these residues, BindMol ensures that the generated ligands delineate the geometric and energetic "fingerprint" of the target pocket. This not only improves affinity but is also crucial for **selectivity**, as engaging a specific constellation of residues helps distinguish the target from homologous proteins.
>
> [1] Enthalpy-entropy compensation in biomolecular interactions. *Chemistry & Biology*, 1996.
>
> [2] Virtual ligand screening: strategies, perspectives and limitations. *Drug Discovery Today*, 2006.
>
> [3] A hot spot of binding energy in a hormone-receptor interface. *Science*, 1995.
>
> > **W5: The Envelope SAC algorithm involves solving an additional optimization (arg max over ω) during training, but the paper does not quantify the resulting computational overhead or its impact on convergence time.**
>
> Thanks for your comments regarding computational efficiency. In practice, we set the sampling size for $\omega$ to $k=5$ to balance performance and efficiency. Furthermore, the preference weights are replicated across the batch dimension. This allows us to perform the preference maximization via parallel tensor computing rather than iterative loops, ensuring high algorithmic efficiency.
>
> To empirically verify that our method does not impose a significant computational burden, we compared the training time of BindMol against several related molecular generation baselines. All experiments were conducted on a single NVIDIA A800 GPU:
>
> | Training Time/hrs        | REINVENT | FREED | MOOD | GEAM | BindMol |
> | ------------------------ | -------- | ----- | ---- | ---- | ------- |
> | Target-based drug design | 2.3      | 9.7   | 11.6 | 12.4 | 11.2    |
>
>  As shown in the results, REINVENT has the highest training efficiency because of its light-weight backbone model (RNN). Within fragment-based generative models, BindMol’s computational cost is comparable (even lower) to other baselines, adding no noticeable overhead. We note that the replication operation on PyG Graph objects does lead to higher memory usage, but this remains well within the accessible computational resources of most research groups (20GB per GPU).

---

> ### Author Response · Authors · 2025-11-21
> **Response to Reviewer Utui (3/4)**
>
> > **W6: In the experimental section, it is not described how the preference vector ω is set during inference, leaving unclear how the model’s multi-objective nature is actually utilized when generating final molecules.**
>
> We thank the reviewer for the careful review and for raising this important question regarding the inference setup. To ensure a strictly fair comparison, we followed the standard evaluation protocols established by FREED [1] and GEAM [2]. Specifically, the molecules used for the final evaluation are collected **continuously during the training process**, rather than being sampled in a separate post-training inference phase. This approach utilizes molecules that have already been evaluated by the oracle/reward function, ensuring that the reported metrics accurately reflect the exploration capability of the model during optimization.
>
> Consequently, there is no need to manually set a fixed preference vector $\omega$ for a separate inference stage, which avoids introducing subjective inductive bias. During the training-collection phase, the weight vector $\omega$ is sampled by first drawing i.i.d. standard normal vectors from $\mathcal{N}(0, I)$, applying an element-wise absolute value, and then performing L1 normalization. This "collect-while-training" strategy ensures that the generated molecules correspond to a diverse range of binding preferences, effectively leveraging the multi-objective nature of our framework to discover novel candidates. We have added this sampling details during training in the `Section 4.2` of revised manuscript.
>
> > **Q2: The model relies on static docking poses to compute residue-level interactions. How do the authors mitigate the risk that suboptimal or strained docking poses lead to spurious interactions being rewarded?**
>
> We fully acknowledge the validity of this concern. It is true that relying on rapid static docking (e.g., QVina) can occasionally result in suboptimal or physically strained poses receiving high scores.
>
> However, we would like to emphasize that this is a **shared limitation across the current landscape of target-based molecular generation**, rather than a specific drawback of our method. All baseline methods compared in this study operate under this standard setting and utilize the same docking-based reward mechanism. By adhering to this protocol, we ensure a strictly fair comparison, verifying that our performance gains stem from the proposed algorithmic framework rather than differences in the scoring quality.
>
> We agree that mitigating this issue is a critical direction for the field. In future work, we aim to address this by incorporating explicit structural constraints and energetic contribution to filter out spurious interactions and ensure physically plausible binding modes.
>
> > **Q3: The dynamic fragment vocabulary is introduced as a novelty. To isolate its contribution, could the authors provide ablation results comparing a fixed-vocabulary variant with identical reward shaping?**
>
> Thanks for your constructive suggestion. We conducted an additional ablation study comparing our full model against a variant using a fixed vocabulary, while keeping all other components (including reward shaping) identical. The comparative results are presented below:
>
> | Novel Hit Ratio           | parp1         | fa7           | 5ht1b         | braf          | jak2          |
> | ------------------------- | ------------- | ------------- | ------------- | ------------- | ------------- |
> | BindMol w/o dynamic vocab | 37.4$\pm$0.79 | 21.6$\pm$0.62 | 39.9$\pm$0.08 | 25.2$\pm$1.67 | 40.1$\pm$1.03 |
> | BindMol                   | 42.5$\pm$0.83 | 24.5$\pm$0.75 | 41.6$\pm$0.14 | 30.9$\pm$1.51 | 44.2$\pm$0.97 |
>
> As shown in the table, restricting the model to a fixed vocabulary leads to a noticeable decline in performance. Moreover, we find that the decline in the novel hit ratio was mainly due to the decrease in the passing rate of novels, while the hit ratio itself did not change significantly.This empirical evidence underscores the critical role of the dynamic vocabulary update mechanism in expanding the searchable chemical space and avoiding local optima.

---

> ### Author Response · Authors · 2025-11-21
> **Response to Reviewer Utui (4/4)**
>
> > **Q4: Multi-objective RL methods often face challenges with instability as the number of objectives grows. The paper cites settings with five to thirty residues. Could the authors report performance as a function of the number of objectives, or provide guidance on stability and hyperparameter sensitivity?**
>
> Thanks for your insightful comments. Regarding the request to plot performance as a function of the number of objectives, we wish to clarify that the number of key residues (and thus the number of objectives) is **intrinsic to the biological target**. Once a target is selected, the objective count is fixed. Consequently, we cannot arbitrarily vary the number of objectives for a single case, and comparing performance across different targets would be confounded by the distinct geometric and chemical properties of different binding pockets.
>
> However, based on our extensive experiments, we can provide **specific empirical guidance regarding stability**. We observed that as the number of objectives increases, the training benefits from a higher ratio of **off-policy updates** (i.e., utilizing more samples from the replay buffer relative to on-policy samples). We hypothesize that in high-dimensional objective spaces, the model requires more frequent re-visitation of past action episodes to effectively learn the complex trade-offs and stabilize convergence.
>
> We have integrated this guidance into the `Appendix B.2` of revised manuscript and highlighted the relevant descriptions for clarity. Thank you again for this constructive suggestion.

---

> ### Author Response · Authors · 2025-11-28
> **Respectful Inquiry Before Discussion Deadline**
>
> Dear Reviewer Utui,
>
> Hope this message finds you well. We wanted to gently remind you that the deadline for the discussion phase is approaching, and we would greatly appreciate it if you could take a moment to review our responses.
>
> Your feedback is very valuable to us, and we wanted to ensure all your concerns have been addressed satisfactorily. If there are any additional concerns or points of clarification, we are more than happy to address them.
>
> Thank you for your time and consideration. We eagerly await your further guidance with utmost respect.
>
> Best wishes,
>
> Authors

---

### Official Review · Reviewer_HeNm · 2025-10-31

**Soundness:** 2
**Presentation:** 3
**Contribution:** 2
**Rating:** 2
**Confidence:** 4

**Summary:**

In this work, the authors introduce BindMol, a novel framework for multi-objective RL in structure-based drug design. Unlike traditional generative models that optimize a single scalar docking score, BindMol decomposes the reward into residue-level objectives, encouraging the generation of molecules with diverse binding modes to the same protein target. The model uses a fragment-based molecular generator and a new RL algorithm called Envelope SAC, which learns a convex envelope over vectorized Q-values to balance multiple interaction objectives efficiently.

Contributions:

Reformulates target-based molecular design as a multi-objective optimization problem focused on residue-level interactions.

Proposes Envelope SAC, a preference-aware RL algorithm for optimizing multiple objectives jointly.

Integrates PLIP-based residue-level rewards with docking scores for richer feedback.

Demonstrates good performance across five protein targets and seven GuacaMol benchmark tasks, with improved hit rates, diversity, and binding affinity compared to prior RL and generative models.

**Strengths:**

Originality:
The paper introduces a conceptually novel formulation of target-based drug design as a multi-objective optimization problem over residue-level interactions rather than a single docking score, and a novel Envelope SAC algorithm.

Quality:
The methodology is technically sound and well-motivated. The empirical validation is in alignment with other studies in molecular generation domain.

Clarity:
The paper is clearly written and logically structured and generally easy to follow.

**Weaknesses:**

The technical novelty of the proposed Envelope SAC algorithm, from the perspective of the ICLR machine learning audience, appears somewhat limited. A more thorough discussion of prior work in multi-objective reinforcement learning including established formulations and optimization strategies would help contextualize the contribution. Furthermore, benchmarking against existing multi-objective RL methods would strengthen the paper’s empirical claims and clarify the specific advantages introduced by the proposed operator. Finally, it would be valuable to know whether the authors have compared their hit rates (Tables 1–3) with recent results such as Pandey et al., “Pretraining Generative Flow Networks with Inexpensive Rewards for Molecular Graph Generation,” arXiv:2503.06337 (2025), which reports strong performance on molecular design tasks for the same targets as considered by the authors. Such comparisons could provide a clearer assessment of BindMol’s relative progress over recent generative paradigms.

**Questions:**

Please see weaknesses

---

> ### Author Response · Authors · 2025-11-21
> **Response to Reviewer HeNm (1/2)**
>
> > **W1: The technical novelty of the proposed Envelope SAC algorithm, from the perspective of the ICLR machine learning audience, appears somewhat limited. A more thorough discussion of prior work in multi-objective reinforcement learning including established formulations and optimization strategies would help contextualize the contribution.**
>
> We thank the reviewer for this constructive feedback. Following your suggestion, we have significantly expanded the discussion of foundational Multi-Objective Reinforcement Learning (MORL) strategies and integrated this into the **Related Work** section of the revised manuscript.
>
> However, we respectfully disagree with the assessment that our technical novelty is limited. We believe our contribution is multifaceted, spanning the **problem definition**, **the application domain**, and **the algorithmic design**:
>
> - To the best of our knowledge, BindMol is the first framework to explicitly target binding mode diversity in molecular generation, and the first to introduce vectorized MORL into this domain. This represents a shift from scalar-based heuristics to a high-dimensional, preference-aware control paradigm.
> - Moreover, implementing the Envelope update within SAC is technically non-trivial. The original Envelope Q-learning was designed strictly for value-based methods. Extending this concept to an Actor-Critic architecture presents unique challenges—specifically, ensuring that both the Actor and the Critic are effectively conditioned on preference vectors and maintain consistent alignment during optimization. Our solution addresses these inconsistencies, going beyond a simple combination of existing components.
>
> For a more detailed elaboration on these points, we kindly invite the reviewer to refer to our `General Response`. We sincerely hope these clarifications address your concerns regarding novelty and warrant a re-evaluation of our work's contribution.
>
> > **W2: Furthermore, benchmarking against existing multi-objective RL methods would strengthen the paper’s empirical claims and clarify the specific advantages introduced by the proposed operator.**
>
> We appreciate the reviewer’s suggestion. As discussed in `Section 5.1` of our manuscript, existing Multi-Objective Reinforcement Learning (MORL) algorithms—typically developed for robotics control or games—generally handle a very limited number of objectives (often fewer than 3). Scaling these methods to the high-dimensional objective space required for our task often leads to a drastic decline in computational efficiency or even makes **training infeasible**. Consequently, the most viable and standard baseline for direct comparison in this setting is the scalarized multi-objective (scalarized MO) approach. We present the performance comparison below:
>
> | Hypervolume   | parp1 | fa7  | 5ht1b | braf | jak2 |
> | ------------- | ----- | ---- | ----- | ---- | ---- |
> | Scalarized MO | 0.24  | 0.43 | 0.19  | 0.16 | 0.18 |
> | BindMol       | 0.32  | 0.51 | 0.20  | 0.19 | 0.30 |
>
> | Novel Hit Ratio | parp1          | fa7            | 5ht1b          | braf           | jak2           |
> | --------------- | -------------- | -------------- | -------------- | -------------- | -------------- |
> | Scalarized MO   | 39.7$\pm$ 0.94 | 18.2$\pm$ 1.03 | 37.8$\pm$ 1.52 | 26.6$\pm$ 0.99 | 40.1$\pm$ 1.18 |
> | BindMol         | 42.5$\pm$ 0.83 | 24.5$\pm$ 0.75 | 41.6$\pm$ 0.14 | 30.9$\pm$ 1.51 | 44.2$\pm$ 0.97 |
>
> As shown in the results, the scalarized multi-objective method achieves a significantly lower Hypervolume and a reduced Novel Hit Ratio. This observation aligns with intuition: by failing to sufficiently explore the Pareto front across diverse preferences, scalarized methods tend to produce repetitive binding modes and structurally similar molecules. This results in **mode collapse** (convergence to local optima), thereby limiting the continuous discovery of novel molecular candidates.
>
> In summary, the distinct advantage of our Envelope SAC algorithm lies in its **scalability to scenarios with more optimization objectives** and its ability to achieve far more effective chemical space exploration compared to scalarized MPO approaches.

---

> ### Author Response · Authors · 2025-11-21
> **Response to Reviewer HeNm (2/2)**
>
> > **W3: Finally, it would be valuable to know whether the authors have compared their hit rates (Tables 1–3) with recent results such as Pandey et al., “Pretraining Generative Flow Networks with Inexpensive Rewards for Molecular Graph Generation,” arXiv:2503.06337 (2025), which reports strong performance on molecular design tasks for the same targets as considered by the authors. Such comparisons could provide a clearer assessment of BindMol’s relative progress over recent generative paradigms.**
>
> We thank the reviewer for bringing this work to our attention. Upon carefully reading, we found that their experimental setting differs from ours and the baselines in Tables 1–3 in two critical aspects:
>
> 1.  **Data Scale:** A-GFN utilizes large-scale pretraining on a dataset of ~6.75M molecules, whereas BindMol are trained **from scratch** without external pretraining data.
> 2.  **Post-processing:** The reported hit discovery performance of A-GFN is partly boosted by a **unique filter** operation, which directly increases the ratio of novel molecules.
>
> To ensure a fair comparison, we evaluated A-GFN using their official code but removed either the large-scale pretraining or the unique filter post-processing, aligning the protocol with our standard benchmarks. The results are as follows:
>
> | Novel Hit Ratio         | parp1         | fa7           | 5ht1b         | braf          | jak2          |
> | ----------------------- | ------------- | ------------- | ------------- | ------------- | ------------- |
> | A-GFN w/o pretraining   | 12.5$\pm$2.39 | 1.7$\pm$1.66  | 20.8$\pm$2.30 | 6.29$\pm$5.13 | 12.3$\pm$1.33 |
> | A-GFN w/o unique filter | 41.7$\pm$0.76 | 6.9$\pm$0.21  | 45.2$\pm$0.12 | 12.1$\pm$3.22 | 47.9$\pm$1.22 |
> | BindMol                 | 42.5$\pm$0.83 | 24.5$\pm$0.75 | 41.6$\pm$0.14 | 30.9$\pm$1.51 | 44.2$\pm$0.97 |
>
> As shown, **BindMol consistently outperforms the train-from-scratch version of A-GFN**. The results also highlight that pretraining is indeed a crucial component for A-GFN’s performance, whereas our method achieves superior results efficiently without reliance on massive external datasets.

---

> ### Author Response · Authors · 2025-11-28
> **Respectful Inquiry Before Discussion Deadline**
>
> Dear Reviewer HeNm,
>
> Hope this message finds you well. We wanted to gently remind you that the deadline for the discussion phase is approaching, and we would greatly appreciate it if you could take a moment to review our responses.
>
> Your feedback is very valuable to us, and we wanted to ensure all your concerns have been addressed satisfactorily. If there are any additional concerns or points of clarification, we are more than happy to address them.
>
> Thank you for your time and consideration. We eagerly await your further guidance with utmost respect.
>
> Best wishes,
>
> Authors

---

### Official Review · Reviewer_zauj · 2025-11-05

**Soundness:** 3
**Presentation:** 3
**Contribution:** 3
**Rating:** 6
**Confidence:** 4

**Summary:**

This paper presents BindMol, a new reinforcement learning framework for structure-based drug design that reformulates molecule generation as a multi-objective optimization problem. Instead of optimizing a single scalar docking score, BindMol defines residue-level rewards based on protein–ligand interaction counts (via PLIP) and introduces Envelope Soft Actor-Critic (Envelope SAC) to explore trade-offs among multiple objectives (binding residues). Experiments on five protein targets and the GuacaMol multi-property benchmark demonstrate that BindMol achieves state-of-the-art performance, generating novel and diverse compounds with improved docking scores and Pareto coverage.

**Strengths:**

- The paper clearly identifies the limitation of scalar docking-based RL and reframes drug discovery as a multi-objective exploration problem, aligning with the biological reality of residue-specific interactions.
- BindMol consistently outperforms a wide range of strong baselines across multiple targets, with substantial gains in both novel hit ratio and binding diversity.
- The use of explorative preferences effectively improves the coverage of chemical space and encourages multiple binding modes, as supported by hypervolume and case study visualizations.

**Weaknesses:**

- The integration with fragment-based generation and residue-level rewards, while well motivated, combines existing ideas rather than introducing a clearly novel modeling mechanism.
- The maximization over ω′ in Equation (8) requires either dense sampling or approximation; the paper does not explain how ω⋆ is computed efficiently or whether it introduces bias.
- The dynamic fragment vocabulary update may leak information from test targets if not strictly separated

**Questions:**

- How does the computational efficiency of BindMol compare to other baseline methods, particularly in terms of training time and docking evaluation cost?
- Since per-residue interaction rewards may incentivize generating larger molecules to form more contacts, how does the molecular weight distribution of samples produced by BindMol compare to those of the baselines?

---

> ### Author Response · Authors · 2025-11-21
> **Response to Reviewer zauj (1/2)**
>
> > **W1: The integration with fragment-based generation and residue-level rewards, while well motivated, combines existing ideas rather than introducing a clearly novel modeling mechanism.**
>
> We appreciate your feedback. We understand the concern regarding the combination of components; however, we respectfully suggest that our contribution goes beyond a simple assembly of existing modules. We have provided a detailed clarification of our conceptual and technical innovations—specifically the Envelope SAC algorithm and the first-class modeling of binding modes—in the **General Response** at the beginning of this rebuttal. We hope these clarifications address your concerns regarding novelty.
>
> > **W2: The maximization over ω′ in Equation (8) requires either dense sampling or approximation; the paper does not explain how ω⋆ is computed efficiently or whether it introduces bias.**
>
> We thank the reviewer for the detailed review and for pointing out this omission. We apologize for not sufficiently elaborating on the implementation details in the original manuscript.
>
> **Sampling Procedure:** The weight vector $\omega$ is sampled by first drawing i.i.d. standard normal vectors from $\mathcal{N}(0, I)$, applying an element-wise absolute value to ensure non-negativity, and then performing L1 normalization. This ensures that the resulting vectors are uniformly distributed on the probability simplex. These weights are then **replicated across the batch dimension** to enable **efficient preference-conditioned training**.
>
> **Efficiency:** Because replicating preference vectors for PyG Graph objects increases memory usage, very dense sampling would significantly increase GPU footprint. In practice, we therefore sample $\omega$ five times (as mentioned in `Appendix B.2`), which provides a good balance between computational efficiency and performance. To further demonstrate the effect of sampling size $k$, we have conducted a sensitivity analysis and the results are shown as follows:
>
> | Novel hit ratio | parp1  | fa7    | 5ht1b  | braf   | jak2   |
> | --------------- | ------ | ------ | ------ | ------ | ------ |
> | k=1             | 35.025 | 18.277 | 36.272 | 19.333 | 39.047 |
> | k=3             | 38.903 | 21.667 | 39.622 | 25.887 | 42.887 |
> | k=5             | 42.538 | 24.471 | 41.639 | 30.867 | 44.238 |
> | k=10            | 43.622 | 24.891 | 42.277 | 31.047 | 45.219 |
> | k=20            | 45.667 | 25.891 | 44.239 | 33.127 | 46.975 |
>
> It can be observed that with samping size increase, the performance of novel hit ratio on three targets steadily increase. However, due to the memory burden, we set sampling size to $k=5$ as the final configuration to achieve the trade-off between performance and efficiency. We have further included this sensitivity analysis in `Table 6` of the revised manuscript and have expanded the description of the sampling mechanism in `Section 4.2`. We sincerely appreciate the reviewer’s suggestion, which helped us clarify this aspect more thoroughly.
>
> > **W3:  Potential information leakage in the dynamic fragment vocabulary update**
>
> Thank you for raising this point. We are not entirely certain what specific form of “information leakage” the reviewer is referring to in this context. The dynamic fragment vocabulary is updated ***only during training***, where docking scores serve as an oracle—consistent with the training setup used by other baseline methods. Importantly, no fragment information from PDB ligands is ever used or exposed in this process. The initial fragment pool is derived solely from molecules in the ZINC250K dataset, and the subsequent updates come exclusively from genetic modifications of these fragments. Therefore, to the best of our understanding, the procedure does not introduce any unfair information leakage.

---

> ### Author Response · Authors · 2025-11-21
> **Response to Reviewer zauj (2/2)**
>
> > **Q1: How does the computational efficiency of BindMol compare to other baseline methods, particularly in terms of training time and docking evaluation cost?**
>
> Thank you for the constructive question and for prompting us to clarify this point. We compared the training times (/hours) of BindMol with several competitive baselines. Since the training time for the same method is basically the same for different targets, we do not distinguish between the targets here; the results are summarized below:
>
> | Training Time/hrs        | REINVENT | FREED | MOOD | GEAM | BindMol |
> | ------------------------ | -------- | ----- | ---- | ---- | ------- |
> | Target-based drug design | 2.3      | 9.7   | 11.6 | 12.4 | 11.2    |
>
> As shown in the results, REINVENT has the highest training efficiency because of its light-weight backbone model (RNN). Within fragment-based generative models, BindMol’s computational cost is comparable (even lower) to other baselines, adding no noticeable overhead. As mentioned in our response to `Weakness 2`, replicating preference vectors across the batch enables efficient parallel computation  rather than loop-based updates, ensuring good computational efficiency.
>
> Regarding docking evaluation cost, we strictly follow prior work so that all baselines, including ours, use the same number of docking calls. Although BindMol performs an additional PLIP-based interaction analysis, the overhead is minimal: with fixed docking poses, PLIP processes $10^5$ molecules within minutes under multi-threading, while our usage is far below this scale. Thus, the added computational cost is negligible. We appreciate the reviewer’s attention to this detail.
>
> > **Q2: Since per-residue interaction rewards may incentivize generating larger molecules to form more contacts, how does the molecular weight distribution of samples produced by BindMol compare to those of the baselines?**
>
> We thank the reviewer for highlighting this interesting and important perspective. Following your suggestion, we compared the molecular weight (MW) distributions of molecules generated by BindMol and by several representative baselines. The results are demonstrated in `Figure 6` of revised manuscript. Additionally, we included the molecular weight of an experimentally validated PDB ligand as references (PDB id = 7F7W).
>
> It  indicates that our per-residue interaction rewards do *not* lead to substantially heavier molecules. The MW distribution of BindMol aligns with those of the baseline methods and is broadly consistent with the distribution observed in PDB ligands. These results indicate that our per-residue interaction rewards do not systematically drive the model toward generating heavier molecules. The similarity between generated molecules and reference ligands further suggests that we operate within a chemically reasonable MW range that is typical for bioactive small molecules.

---

> ### Comment · Reviewer_zauj · 2025-11-26
>
> I really appreciate the authors for their detailed rebuttal responses. I believe this will be a good paper for this field. I will keep my positive score

---

> > ### Author Response · Authors · 2025-11-27
> >
> > Thank you very much for your thoughtful evaluation and encouraging comments! We sincerely appreciate your support and are grateful for your positive assessment.

---

### Author Response · Authors · 2025-11-21
**General Response**

Dear Reviewers and Chairs,

We would like to thank all reviewers very much for their extensive reviews and constructive critiques, which are invaluable for improving our manuscript. We are encouraged that reviewers highlight several positive aspects of our work, including the conceptual novelty of our problem formulation (Reviewers `zauj`, `HeNm` and `dzp3`), the breadth and competitiveness of our empirical evaluation (Reviewers `Utui`, `dzp3` and `zauj`), and the technical contribution of our proposed RL algorithm (Reviewer `Utui`).

However, we notice that the novelty and contribution have not been fully captured, leading to potential misunderstandings of some reviewers. We therefore take this opportunity to clearly articulate the core innovations of **BindMol**.

 **(1) Novel problem formulation.**
To the best of our knowledge, BindMol is the first molecular generation framework that explicitly accounts for the *diverse binding modes between ligands and a target*. This is a biologically grounded phenomenon repeatedly validated in drug discovery practice, where ligands with distinct binding poses may exhibit differentiated functional, selectivity, and developability profiles [1]. Existing molecular generation approaches typically overlook this intrinsic diversity and instead assume a single binding configuration. By contrast, **our formulation treats binding-mode diversity as a *first-class modeling objective***, which opens up a new direction for structure-based molecule design. We believe this conceptual shift itself constitutes a significant and underexplored innovation.

[1] Differential glp-1r binding and activation by peptide and non-peptide agonists. Molecular Cell, 2020.

**(2) First integretion of vectorized MORL in molecular generation.**
 Current multi-objective molecular generation approaches predominantly rely on Bayesian optimization or on scalarization heuristics that reduce multiple objectives to a single weighted score. Our work is, to our knowledge, the first to explore a principled integration of *vectorized multi-objective preference signals directly within an RL framework* for molecular generation. This enables fine-grained preference-aware optimization and avoids the limitations of fixed scalarization, thereby introducing a new paradigm for controllable multi-objective molecule design.

**(3) Technical novelty of the Envelope SAC algorithm.**
 We would also like to clarify the misunderstanding that Envelope SAC is merely a combination of the envelope update and SAC. The original envelope operator was developed in the context of value-based deep Q-learning, where the policy is implicit and no explicit actor is optimized. Extending this concept to an actor–critic setting is fundamentally non-trivial: both the critic and the policy must be preference-aware, and naïve adaptation leads to inconsistent optimization objectives. Our approach proposes **a unified framework that jointly optimizes *multi-preference-aware critics and actors*, and introduces an exploration mechanism tailored for preference diversity.**

We acknowledge that our initial writing placed substantial emphasis on prior work that inspired certain components, which may have unintentionally conveyed an impression of incremental improvement. In the `Section 1` of revised version, we have refined and highlighted the specific conceptual and technical contributions to avoid such ambiguity.

Finally, we appreciate all your helpful comments that strengthen the quality and clarity of our work. We also provide detailed point-to-point response to other concerns and questions in the following responses. And we look forward to engaging in an active and productive discussion with the reviewers. If you have any other questions, please feel free to let us know.

Best regards,

The Authors

---

> ### Author Response · Authors · 2025-12-02
> **Rebuttal Summary**
>
> Dear Chairs,
>
> We are writing to provide a consolidated summary of our rebuttal, particularly in light of the system interruption that prevented a final dialogue with the reviewers.
>
> **1. Summary of Reviewer Positions**
>
> Overall, the feedback presents a mix of positive recognition and constructive critique. Reviewers `zauj` were largely positive, endorsing the conceptual value and empirical success of our work. Reviewer `Utui` raised valid questions regarding theoretical clarity and biochemical rationale. Reviewer `HeNm` and `dzp3` expressed some concerns, primarily centering on novelty.
>
> **2. Consensus on Strengths**
>
> We are encouraged that the reviewers collectively recognized the **conceptual novelty** of our problem formulation—specifically, reframing drug discovery to explicitly model binding-mode diversity. They also acknowledged the **strong empirical performance** of BindMol across multiple biological targets and the soundness of our experimental validation.
>
> **3. Response to Major Concerns**
>
> In our `General Response`, we have comprehensively addressed the primary concerns regarding novelty. We clarified that our innovation is multifaceted—spanning the **conceptual formulation** of binding-mode diversity, the **first integration** of vectorized MORL in molecular generation, and the **algorithmic design** of Envelope SAC—rather than a mere combination of existing components.
>
> Furthermore, we have made a significant effort to address every point raised, including conducting extensive additional experiments (sensitivity analysis, runtime comparisons, more baseline comparisons, and ablation studies) during the short rebuttal window. Although the discussion period was unfortunately curtailed, we believe our rebuttal has technically and empirically resolved the reviewers' concerns.
>
> We remain respectful of the review process and trust your judgment in weighing these points during the discussion stage. We hope this context will assist in forming a balanced final decision.
>
>
>
> Sincerely,
>
> The Authors

---

### Meta-Review · Area_Chair_zFRV · 2026-01-07

**Summary:**

Based on the review discussion, the paper presents BindMol, a framework that reformulates structure-based drug design as multi-objective reinforcement learning over residue-level interactions rather than scalar docking scores. The reviewers acknowledged several strengths including the conceptual novelty of modeling binding-mode diversity explicitly, strong empirical performance across multiple benchmarks, and the technical contribution of the Envelope SAC algorithm.

However, concerns about novelty emerged as a central theme across multiple reviews. Reviewer dzp3 questioned whether the work represented sufficient innovation given that it combines existing components like FREED's action space with envelope Q-learning concepts from prior MORL work. Reviewer HeNm raised similar concerns about limited technical novelty from a machine learning perspective and suggested more thorough contextualization within the broader MORL literature. Reviewer Utui pointed out ambiguities in how scalar rewards integrate with the multi-objective formulation and questioned the biochemical justification for why interacting with more residues should necessarily improve binding quality.

The authors provided extensive rebuttals including additional experiments on sensitivity analysis, computational efficiency comparisons, molecular weight distributions, and ablations on dynamic vocabulary and scalarized baselines. They argued that extending envelope updates from value-based to actor-critic settings is non-trivial and that their explicit modeling of binding modes represents a conceptual paradigm shift. Reviewer zauj responded positively to the rebuttal and maintained their acceptance recommendation. However, the other three reviewers did not engage further during the discussion phase despite author reminders, leaving their concerns regarding novelty and theoretical clarity unresolved. This lack of reviewer engagement combined with the initial reservations about contribution creates uncertainty about whether the authors' clarifications adequately addressed the fundamental concerns raised during the review process.

**Reviewer Concerns:**

The authors provided comprehensive rebuttals that addressed several reviewer concerns effectively, though some fundamental issues remained partially unresolved due to lack of reviewer re-engagement.

Concerns that were adequately addressed include Reviewer zauj's questions about computational efficiency where the authors provided detailed training time comparisons showing BindMol requires comparable resources to other fragment-based methods around 11 hours versus 9-12 hours for baselines. The molecular weight distribution analysis demonstrated that residue-level rewards do not systematically bias toward heavier molecules with distributions aligning closely to PDB reference ligands. The sensitivity analysis on preference sampling size k showed performance gains plateauing around k equals 5 which justified their configuration choice balancing efficiency and performance.

Reviewer Utui's concern about the connection between scalar and vectorized rewards was clarified through corrections to notation explaining that residue-specific rewards correspond to vector components while docking scores provide global signals. The ablation study on dynamic vocabulary showed meaningful performance drops of 3-5 percentage points when using fixed vocabularies validating this component's contribution. Computational overhead concerns were addressed with the training time analysis and explanation of parallel tensor operations rather than iterative loops.

For Reviewer HeNm, the comparison against scalarized multi-objective baselines demonstrated clear advantages in both hypervolume and novel hit ratios. The adjusted comparison with A-GFN after removing pretraining or unique filtering showed BindMol's superiority under fair conditions without massive external datasets.

However, several concerns remain outstanding primarily because reviewers did not re-engage after rebuttals. Reviewer dzp3's fundamental critique about weak novelty persists since the argument that Envelope SAC represents non-trivial extension from value-based to actor-critic methods may not be fully convincing without explicit theoretical analysis or architectural diagrams showing the coordination challenges. The claim of paradigm shift in treating binding-mode diversity as first-class objective versus existing scalar optimization remains somewhat philosophical rather than technically grounded.

Reviewer HeNm's concern about limited technical novelty from machine learning perspective was not fully resolved despite expanded MORL discussion since the core question remains whether the adaptation constitutes sufficient algorithmic innovation for a top-tier ML venue. The missing comparisons with other established MORL algorithms beyond scalarized baselines leaves incomplete empirical validation of the specific advantages of Envelope SAC.

Reviewer Utui's biochemical rationale question about why more residue interactions should improve affinity received thermodynamic arguments citing enthalpy-entropy compensation and hotspot theory but these feel somewhat post-hoc rather than demonstrating that the reward design causally drives the observed performance gains. The equal weighting of interaction types acknowledged as limitation suggests the reward function may be capturing correlation rather than mechanistic understanding. The reliance on static docking poses as shared limitation across the field does not address whether spurious interactions might disproportionately affect multi-objective optimization compared to scalar approaches.

**Reviewer Scores:**

Reviewer zauj initially scored 6 and explicitly stated after the rebuttal "I really appreciate the authors for their detailed rebuttal responses. I believe this will be a good paper for this field. I will keep my positive score." This reviewer was satisfied with the authors' responses on computational efficiency, molecular weight distributions, and sensitivity analysis. The score would likely remain at 6.

Reviewer HeNm initially scored 2 and raised concerns about limited technical novelty and missing comparisons with recent methods like the Pandey et al. pretraining work. The authors provided comparisons showing BindMol outperforms A-GFN without pretraining and explained the unfair advantage of large-scale pretraining in those baselines. They also compared against scalarized multi-objective methods demonstrating clear advantages. However, the fundamental concern about whether Envelope SAC represents sufficient algorithmic contribution for ICLR's machine learning audience was not deeply engaged. The expanded MORL discussion and clarifications about actor-critic challenges might partially address this but the reviewer's silence suggests either insufficient time for re-evaluation or persistent skepticism. The score might cautiously increase to 4 if the reviewer found the additional experiments and contextualization adequate.

Reviewer Utui initially scored 4 and provided the most technically detailed review with concerns about theoretical clarity, biochemical rationale, and computational overhead. The authors systematically addressed notation ambiguities, provided training time comparisons, added ablation studies on dynamic vocabulary, and offered thermodynamic arguments for the residue-interaction design. The comprehensive point-by-point responses with additional experiments demonstrate good faith effort to address every concern raised. However, some issues like equal weighting of interactions and reliance on static docking remain acknowledged limitations rather than resolved problems. Given this reviewer was already marginally below acceptance at 4 and the authors provided substantial additional evidence and clarifications, the score would likely increase to representing a shift to marginally above acceptance threshold, though uncertainty remains without direct reviewer confirmation.

Reviewer dzp3 initially scored 4 with the primary weakness being concerns about weak novelty since BindMol combines existing components like FREED's action space with MORL concepts. The authors argued that extending envelope updates to actor-critic settings is non-trivial and that explicit binding-mode modeling represents conceptual innovation. However, this reviewer's concern was fundamentally about whether the integration constitutes sufficient contribution rather than technical correctness. The missing comparisons with GenMol and Genetic GFN were explained as unfair due to pretraining advantages but this reasoning might not fully satisfy a reviewer who views comprehensive empirical validation as essential. Without re-engagement, it is difficult to assess whether the novelty arguments would be persuasive. The score might increase to 6 if the reviewer accepted the paradigm shift framing, but could remain at 4 if the incremental nature concern persisted since the core architectural choices remain borrowed from prior work.

---

### Decision · Program_Chairs · 2026-01-26

Reject